# Performance assessment of high-density diffuse optical topography regarding source-detector array topology

**Hadi Borjkhani, Seyed Kamaledin Setarehdan** *

Control and Intelligent Processing Centre of Excellence, School of Electrical and Computer Engineering, College of Engineering, University of Tehran, Tehran, Iran

* ksetareh@ut.ac.ir

## Abstract

Recent advances in optical neuroimaging systems as a functional interface enhance our understanding of neuronal activity in the brain. High density diffuse optical topography (HD-DOT) uses multi-distance overlapped channels to improve the spatial resolution of images comparable to functional magnetic resonance imaging (fMRI). The topology of the source and detector (SD) array directly impacts the quality of the hemodynamic reconstruction in HD-DOT imaging modality. In this work, the effect of different SD configurations on the quality of cerebral hemodynamic recovery is investigated by presenting a simulation setup based on the analytical approach. Given that the SD arrangement determines the elements of the Jacobian matrix, we conclude that the more individual components in this matrix, the better the retrieval quality. The results demonstrate that the multi-distance multi-directional (MDMD) arrangement produces more unique elements in the Jacobian array. Consequently, the inverse problem can accurately retrieve the brain activity of diffuse optical topography data.

## Introduction

Recent developments in functional neuroimaging systems quantitively enhance our understanding of spatially and temporally distributed neural activity in the brain [1,2]. Functional near-infrared spectroscopy (fNIRS) is a new, emerging, and growing technology for monitoring neurological activity in which red and near-infrared light is used to measure changes in Oxy- and Deoxyhemoglobin in brain tissue [3–6]. The fNIRS is an optical neuroimaging technology that is radiation-free, relatively inexpensive, compatible with implanted electronic devices, and portable and can wirelessly record brain activity [7–10]. Although the fNIRS systems are mobile and compact, the resolution and depth of the imaging are less than those obtained by the fMRI [11]. HD-DOT allows brain activity to be mapped in 3D by creating overlaps between the fNIRS channels. This method uses high-density SD arrays to improve the spatial resolution that is comparable to fMRI [10].

SD configuration and number of overlapped channels alongside the inverse problem significantly influences the spatial resolution [10,12]. In this paper, the impact of SD topology on the spatial resolution and hemodynamic reconstruction of HD-DOT has been investigated. We

**Data Availability Statement:** All relevant data are within the manuscript and Supporting Information files.

**Funding:** The author(s) received no specific funding for this work.

**Competing interests:** The authors have declared that no competing interests exist.

have developed an analytical simulation setup to evaluate the performance of the different combinations of SD on hemodynamic regeneration. Also, this simulation setup can be employed to optimize the arrangement of SD and the number of multi-distance channels and performance of the inverse problem on hemodynamic reconstruction. The SD arrangement and the number of channels determine the elements of the Jacobian matrix. We observed that the MDMD arrangement produces more unique components in the Jacobian matrix. The outcomes of this work indicate that the more individual ingredients in this matrix, the better the reconstruction quality. The Jacobian is the sensitivity matrix, which is computed by the forward model. In this work, the solution of Diffusion Equation inside inhomogeneous media similar to properties of the brain tissue constitutes the base of the forward model.

The forward model is a part of the simulation setup, which plays a fundamental role in confirming the results of this research. Therefore, in the discussion section, the performance of the forward model [13] used in this study is compared with the statistical model on Colin 27 brain template [14].

The rest of the study has continued as follows. The second section describes the simulation setup, which is applied to four different topologies of SD on the forward model. In the following, the synthetic fNIRS data are modeled and generated to simulate hemodynamic changes in the brain. Finally, based on the calculated diffuse reflectance, the inverse method was employed for hemodynamic reconstruction. Part 3 represents the simulation of the forward model and hemodynamic reconstruction. The discussion and conclusion of this study are outlined in sections 4 and 5.

## Materials and methods

### Modeling approach

The block diagram of the proposed simulation setup illustrated in Fig 1 contains all the steps taken in this article. This scheme employs an analytical forward model that has less computation time. As the number of channels increases, the computational volume in the modeling increases, so the use of analytical models are preferred to numerical approaches [15,16]. Since the perturbative Diffusion Equation (pDE) equations do not have an analytical solution in complex geometries, the simple geometry, that can approximate a semi-infinite medium for thick slabs, is utilized [16].

The sources and detectors are aligned in XY-plane on the top of one layer slab geometry. Synthetic cerebral hemodynamic is generated to model more realistic reflectance. The cost function, which is used to modify the regularization parameter, is the correlation between reconstructed and synthetic data. We will prove that the accuracy of reconstruction depends on the SD arrangement, the number of channels, and the energy regularization parameter. Here the forward model plays an essential role since the validation of a model to realistic results mainly depends on it.

Also, the simulation setup is expandable for many quantities of perturbation inside the medium and can be used to evaluate the performance of HD-DOT. The purpose of this scheme is to recover the synthetic hemodynamic in location S1-S9, particularly the recovery of S5. The different arrangement of SD is applied to the simulation setup, and the potential of each combination is analyzed in hemodynamic reconstruction. The first arrangement (Fig 1 (a)) represents one repeatable part of HD-DOT [10]; the other remaining provisions are proposed to view the effect of different SD topology on hemodynamic reconstruction (Fig 1(b), 1 (c) and 1(d)). The block diagram of Fig 1(e) illustrates the analytical simulation setup. The following sections will describe this diagram in detail.

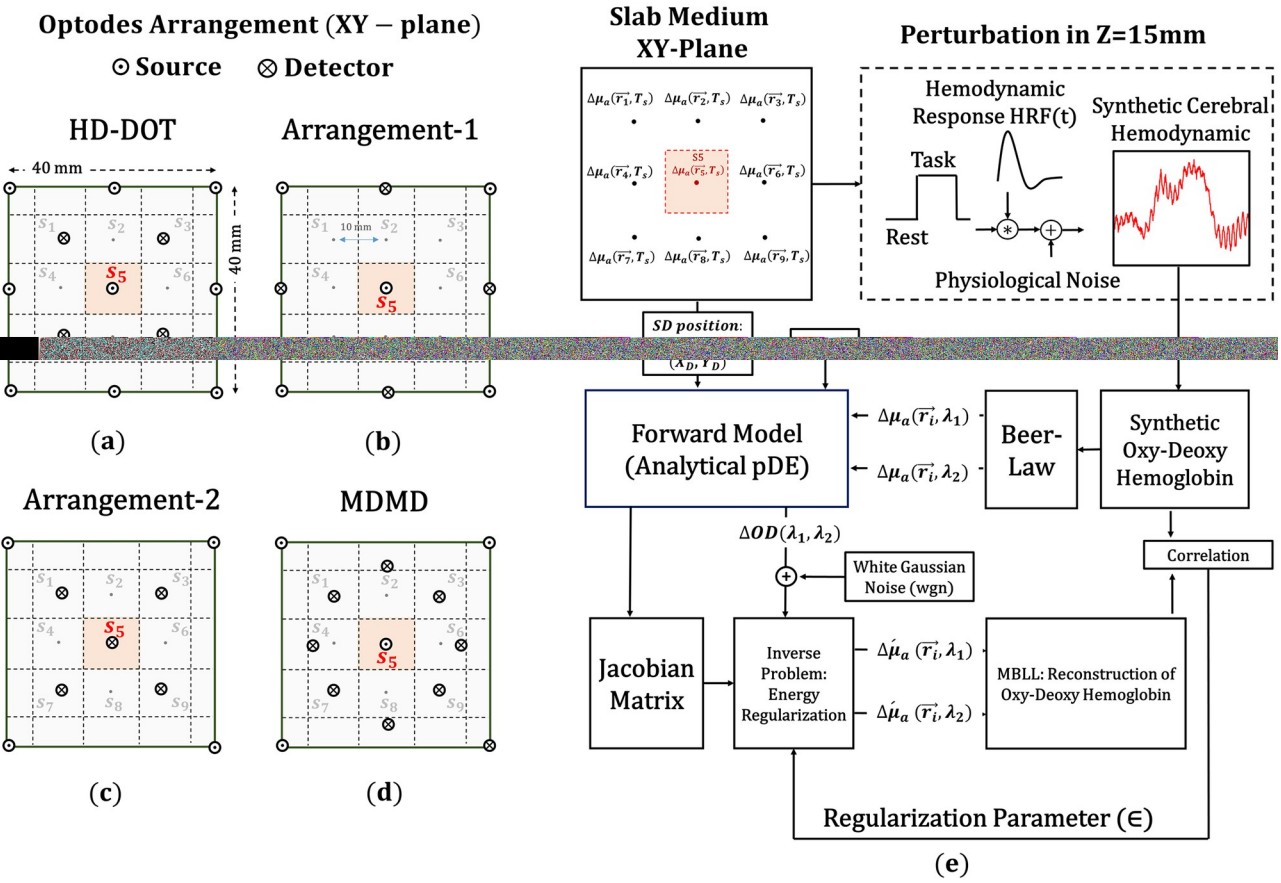

**Fig 1. Illustrates the modeling approach in this work and the different arrangements of source and detector in XY-plane.** (a), (b), (c), and (d) represent the configuration of the SD in four different modes (HD-DOT, Arrangement-1, Arrangement-2, and MDMD topology). (e) Block diagram of analytical simulation setup.

## Arrangement of SD

There is a direct relationship between SD topology and the accuracy of hemodynamic signal reconstruction. The higher the number of channels with different overlapped directions and distances, the higher the efficiency of the recovery. For the same amount of SD, MDMD arrangement (shown in Fig 1(d)), has higher multi-distance, multi-directional channels compared to HD-DOT. Arrangement-1 and arrangement-2 both have the same number of SD, but the lowest quantity of channels belongs to arrangement-1. These four arrangements are illustrated in Fig 2 and compared in terms of multi-distance, multi-directional, and the number of channels.

The simulation setup was created based on the analytical solution of perturbation theory to verify the accuracy of the SD arrangement in the reconstruction of the hemodynamic response, which will be described in the next section.

## Theory for reflectance perturbation

The analytical solution for perturbative DE has been obtained for the geometry of Fig 3. The geometry of the boundary for the analytical solution of the perturbative DE is a slab. The slab geometry is widely used for calculation of the reflectance in brain functional imaging [16–20]. It is better to note that the boundary is not limited in the direction of axis X and Y [16].

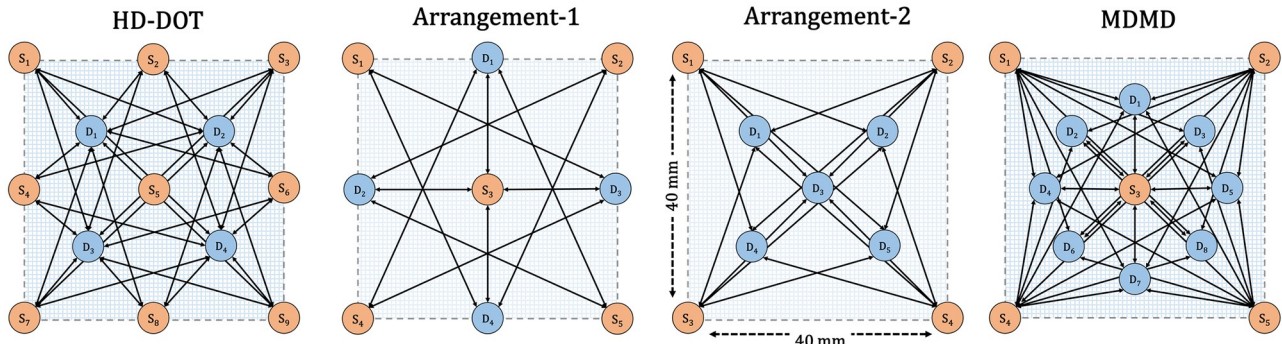

**Fig 2. (a), (b), (c), and (d) indicate the SD location, number of channels, number of distinct directions, and distance in XY-plane for HD-DOT, arrangement-1, arrangement-2, and MDMD respectively.**

For the sake of simplicity, the geometry has been supposed to have only one layer. The absorption and scattering coefficient of the slab respectively considered to be $\mu_a = 0.01mm^{-1}$ and $\mu'_s = 1mm^{-1}$ and the thickness of slab is equal to40mm and refractive index $n_r = 1.4$ [16].

The result obtained by this approach is accurate when the defect causes small perturbation on photon migration. Consequently, the volume of the inhomogeneity (inclusion) is regarded to be small concerning baseline optical properties of the homogeneous medium [16].

The reflectance of each channel has been calculated for several inclusions inside the medium. Reflectance in inhomogeneous media is the superposition of the reflectance inside homogeneous media $R^0(\rho, t)$, plus the absorption ($\delta R^a(\rho, t)$) and scattering ($\delta R^D(\rho, t)$) effect of inclusion [16].

$$R^{pert}(\rho, t) = R^0(\rho, t) + \delta R^a(\rho, t) + \delta R^D(\rho, t) \tag{1}$$

Inside the slab medium, nine dynamic perturbations are inserted to simulate the real function of the brain in the cerebral cortex. Each perturbation is located at the center of a voxel at $15mm$ depth.

The final expression of $R^{pert}(\rho)$ For each channel derived based on Born approximation [21]. Finally, the diffuse reflectance for the channel between a source located in $S_x$ and a detector at $D_x$ would be:

$$R^{pert}(\rho_j, T_s) = R^0(\rho_j) + \sum_{i=1}^{9} \delta R^a(\rho_j, T_s, i) \tag{2}$$

Where $R^0(\rho_j)$ is the reflectance for homogeneous media and the $\delta R^a(\rho_j, T_s, i)$ is the absorption perturbation of $i^{th}$ inclusion and $T_s$ is the sampling time of dynamic perturbation.

The channel definition in Eq (2) is based on the source ($S_x$) and detector ($D_x$), and the corresponding distance between them ($\rho_j$).

The position of SD in Cartesian coordinates on the surface of Slab geometry (XY-plane) and nine perturbed inclusion inside it has displayed in Fig 3. For example, the dynamic perturbation located at the center of $S_9$ voxel has shown in this figure. The distance between adjacent voxels is considered to be 10mm. Next section will describe how this dynamic perturbation is generated.

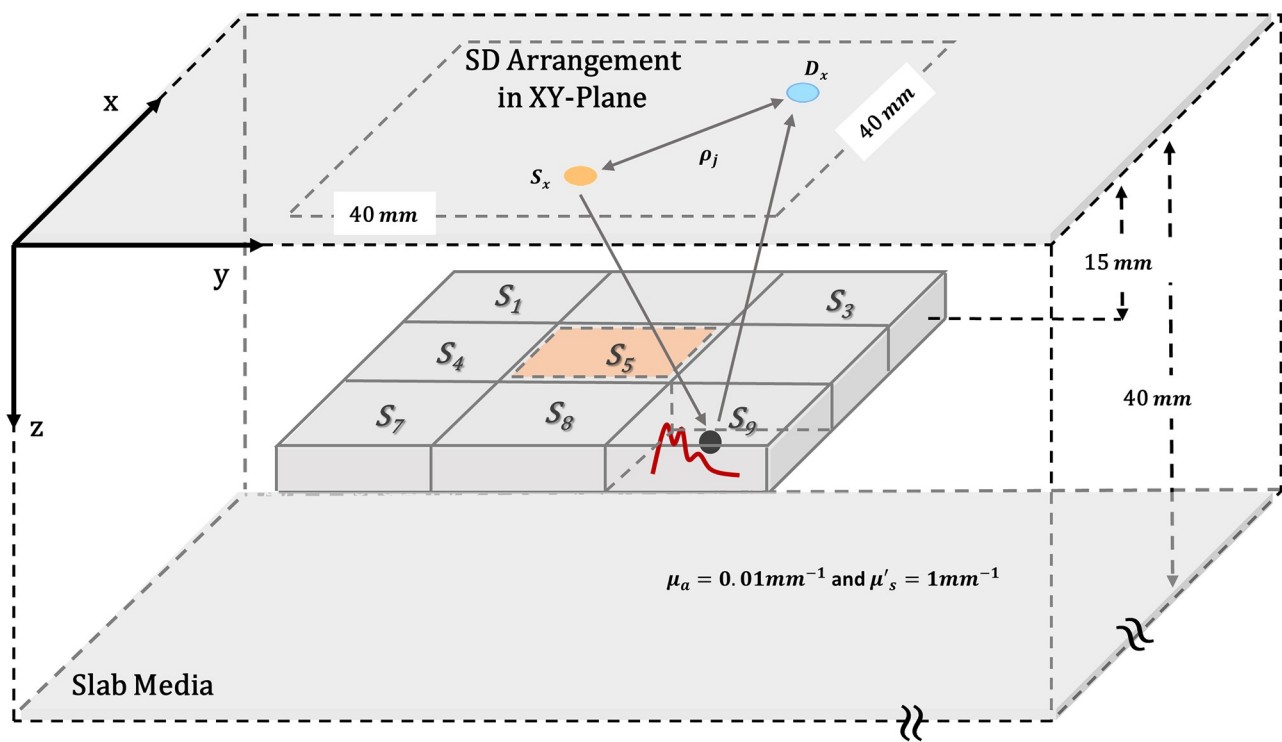

**Fig 3. Location of SD in Cartesian coordinates on the surface of slab geometry and nine perturbed inclusion inside it.**

### Synthetic fNIRS data

This section describes the scheme for simulating the perturbation inside the medium. So $R^{pert}(\rho_j, T_s)$ is modulated using the synthetic Hemodynamic Response Function (HRF). The $\Delta HbO_2$ and $\Delta Hb$ concentration in this medium were perturbed in nine regions, to mimic hemodynamic response concerning the duration of the task. The event or task duration was considered to be random to examine the performance of the inverse problem in all possible states. The perturbation is generated by the convolution of the boxcar function with synthetic hemodynamic. Boxcar function (s(t)) is regularly repeated as a rectangular pulse waveform with modulated duty cycle (related to the duration of task). The amplitude of 1 indicates the task, and 0 refers to rest [22]. S(t) is the pulse-width modulated signal:

$$s(t) = \begin{cases} 1 & t_i : t_i + d_j \\ 0 & otherwise \end{cases} \tag{3}$$

Where, $i = 1: N$; $j = 1,2, \ldots, (N - t_i)$, and $d_j$ represents the duration of each task. The HRF(t) was modeled as a linear combination of two different gamma variant time-dependent function [23]:

$$HRF(t) = \alpha \times [\varphi(t, \tau_1, \rho_1) - \beta \times \varphi(t, \tau_2, \rho_2)] \tag{4}$$

With:

$$\varphi\left(t, \tau_j, \rho_j\right) = \frac{1}{p!\tau_j} \left(\frac{t - \rho_j}{\tau_j}\right)^p e^{\frac{-(t-\rho_j)}{\tau_j}} u\left(t - \rho_j\right), \quad u\left(t - \rho_j\right) = \begin{cases} 1 & if(t - \rho_j) \geq 0 \\ 0 & otherwise \end{cases} \tag{5}$$

Where $\alpha$ determines the amplitude, $\rho_1$, and $\rho_2$ regulate the starting, end, and duration of HRF, $\tau_1$, and $\tau_2$ tune the ascending and descending shape of HRF, while $\beta$ control the undershoot. The value of $p$ coefficient recommended being five [24]. The HRF profile with a peak amplitude of almost 1555 nM was chosen for $HbO_2$ while the Hb profile is the same as HRF for $HbO_2$ but with an inverted magnitude by 33% attenuation and regulated latency [23]. The change of $HbO_2$ corresponding to each perturbation would be the convolution of the $HRF_i(t)$ and $s(t)$ plus physiological noise:

$$\Delta HbO_{2_i}(t) = HRF_i(t)*s(t) + \emptyset_{phy}(t) \; and \; \Delta Hb_i(t) = -\frac{1}{3} \times \Delta HbO_{2_i}(t) \tag{6}$$

The physiological noise was modeled as a linear combination of sinusoids [25]:

$$\emptyset_{phy}(t) \;\; = \;\; \sum_{i \; = \; 1}^{5} [A_i \; \sin(2\pi f_i t + \theta_i)] \tag{7}$$

The $\emptyset_{phy}(t)$ for each perturbation is the average of the ten trials of the Eq (7). The value of amplitude and frequency of the sinusoids would be different for each repetition, while phase $\theta_i$ are equally distributed between 0 and $2\pi$ for each trial [23].

## Inverse problem: Hemodynamic reconstruction

The detectors record the variation in the light intensity, which is formed by the corresponding source. These detectors represent the optical properties of the channel that is called optical density. The Modified Beer-Lambert law (MBLL) is used to relate changes in optical density to changes in concentration of Oxy and Deoxyhemoglobin under the assumption that the scattering losses are constant (S1 Appendix indicates further information about these equations). The variation in Oxy-Hemoglobin ($\Delta HbO_2(t)$) and Deoxy-Hemoglobin ($\Delta Hb(t)$) according to Beer law [26] determines the change in absorption coefficient ($\Delta\mu_a(\lambda)$). According to Beer's law [26]:

$$\Delta\mu_a = \sum_{i=1}^{n} \varepsilon_i \Delta c_i \tag{8}$$

Where, n represent the number of the light absorbing agent (chromophores) in the tissue. However, in near-infrared wavelength (700-900nm), the dominant absorption changes are caused by the concentration of $O_2Hb$ and HHb. As a result, the $\Delta\mu_a$ can be expressed by:

$$\Delta\mu_a(\lambda) = \varepsilon_\lambda^{HbO_2}.\Delta HbO_2 + \varepsilon_\lambda^{Hb}.\Delta Hb \tag{9}$$

Where, $\varepsilon_\lambda^{HbO_2}$ and $\varepsilon_\lambda^{Hb}$ represent the extinction coefficients of Oxy and Deoxyhemoglobin, respectively. The forward model is formed by using the synthetic fNIRS and perturbation theory in two wavelengths.

The solution of the inverse problem to the forward model is required to estimate the synthetic hemodynamic. The forward model can be rewritten as:

$$\Delta R_j^{pert}(T_s)/ R_i^0(\rho_j) = J(\mu_a(\vec{r}))_{j\times i}\Delta\mu_a(\vec{r_i}, T_s) \tag{10}$$

Where, $\Delta\mu_a(\vec{r_i}, T_s)$ is the absorption perturbation for each location of the domain under the head surface. $\Delta\mu_a(\vec{r_i}, T_s)$ is modulated by synthetic hemodynamic and $J(\mu_a(\vec{r}))$ is the Jacobian

matrix (it shows the sensitivity of reflectance to each perturbation in specific depth):

$$J(\mu_a(\vec{r}))_{j \times i} = \begin{pmatrix} \delta R^a(\rho_1)/R_1^0(\rho_1) & \delta R^a(\rho_1)/R_2^0(\rho_1) & \dots & \delta R^a(\rho_1)/R_i^0(\rho_1) \\ \delta R^a(\rho_2)/R_1^0(\rho_2) & \delta R^a(\rho_2)/R_2^0(\rho_2) & \dots & \delta R^a(\rho_2)/R_i^0(\rho_2) \\ & & \vdots & \\ \delta R^a(\rho_j)/R_1^0(\rho_j) & \delta R^a(\rho_j)/R_2^0(\rho_j) & \dots & \delta R^a(\rho_j)/R_i^0(\rho_j) \end{pmatrix} \quad (11)$$

Where index "j" refers to the number of channels and index "i" refers to the number of perturbations under the SD array. There are several approaches to solve the inverse problem of Eq (10) [27]. One of the commonly employed methods to provide a solution to the inverse problem is energy regularization [28]. Reconstruction of the hemodynamic is obtained by:

$$\Delta\mu_a(\vec{r}, T_s) = J(\mu_a(\vec{r}))^T [J(\mu_a(\vec{r}))J(\mu_a(\vec{r}))^T + \epsilon I]^{-1} \Delta R^{pert}(T_s) \quad (12)$$

Where, $J(\mu_a(\vec{r}))^T$ is the transposition of the Jacobian matrix, "$\epsilon$" is the energy regularization parameter and "I" is the identity matrix. The optimum value for "$\epsilon$" is found empirically based on the simulation results.

Given the Eq (9), $\varepsilon_\lambda^{HbO_2}$ and $\varepsilon_\lambda^{Hb}$ are a function of the wavelength. By calculating the variation of optical density $\Delta OD$ at two wavelengths (in this paper 780 and 820 nm), and assuming that the length of the traveling light is identical in both wavelengths, the values of $\Delta\mu_a(\lambda_1)$ and $\Delta\mu_a(\lambda_2)$ are obtained. As a result, the corresponding hemodynamic can be reconstructed:

$$\Delta(HbO_2) = \frac{\varepsilon_{\lambda2}^{Hb} \cdot \Delta\mu_a(\lambda_1) - \varepsilon_{\lambda1}^{Hb} \cdot \Delta\mu_a(\lambda_2)}{\varepsilon_{\lambda1}^{HbO_2} \times \varepsilon_{\lambda2}^{Hb} - \varepsilon_{\lambda1}^{Hb} \times \varepsilon_{\lambda2}^{HbO_2}} \quad (13)$$

$$\Delta(Hb) = \frac{\varepsilon_{\lambda2}^{HbO_2} \cdot \Delta\mu_a(\lambda_1) - \varepsilon_{\lambda1}^{HbO_2} \cdot \Delta\mu_a(\lambda_2)}{\varepsilon_{\lambda1}^{Hb} \times \varepsilon_{\lambda2}^{HbO_2} - \varepsilon_{\lambda1}^{HbO_2} \times \varepsilon_{\lambda2}^{Hb}} \quad (14)$$

Where in this work $\lambda_1 = 780nm$ and $\lambda_2 = 820nm$. The extinction coefficients of Oxy and Deoxy hemoglobin for both wavelengths are given by [29].

## Results

### Forward simulation

The elements of the Jacobian matrix for the given arrangement have been calculated by sweeping one inhomogeneity in a 3D position in the medium under study. The contrasts ($\delta R^a(\rho_j)/R^0(\rho_j)$) of channels (Fig 4) have been obtained for 10mm, 20mm, 30mm, 40mm, 50mm, and 60mm distance between SD. The contrast indicates the sensitivity of reflectance for any perturbation inside 3D geometry. This simulation result shows less sensitivity in profound depth. This feature has been described comprehensively and in detail by [27]. According to this figure, penetration depth increases throughout the growing distance among SD.

The synthetic oxyhemoglobin ($\Delta(HbO_2)$) and deoxyhemoglobin ($\Delta(Hb)$) are generated with the corresponding $\Delta\mu_a(\lambda)$ at the wavelength of the 780nm and 820nm (Fig 8). The solution of DE based on perturbation theory and Born approximation for nine perturbations is employed. The transient brain activity is modeled in a way to modulate absorption. So $\Delta\mu_a(\lambda)$ can be time-dependent ($\Delta\mu_a(\lambda, t)$). Then the reflectance due to perturbations inside the medium depends on $\Delta\mu_a(\lambda, t)$ of each perturbation. Consequently, $R^{pert}(\rho, T_s)$ is modulated using synthetic hemodynamic.

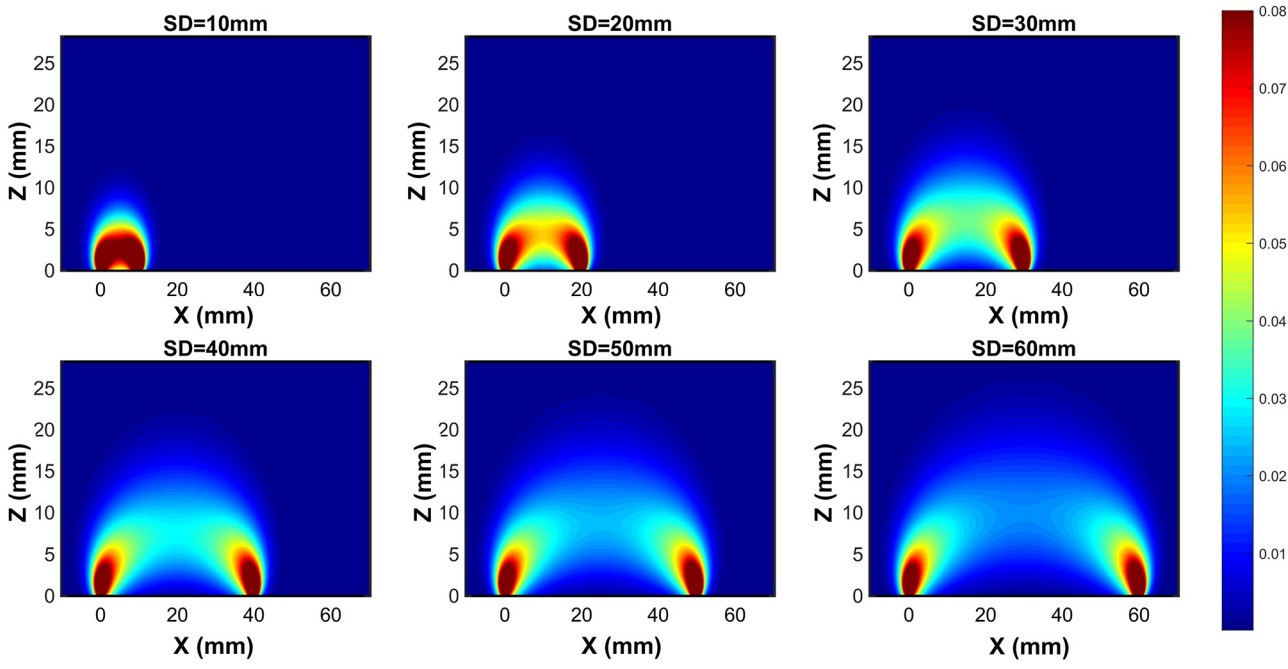

**Fig 4. The contrast ($\delta R^a(\rho_j)/R^0(\rho_j)$) in XZ-plane for SD separation from 10mm to 60mm.**

According to the forward model, nine synthetic hemodynamic response is generated at 15$mm$ depth and reconstructed by the solution of the inverse problem. Fig 5 shows the change in synthetic oxy and deoxyhemoglobin. This figure represents that each hemodynamic has a distinct pattern compared to others. The different hemodynamic trends in each voxel make it more challenging to recover hemodynamics, and under these conditions, the capability of the

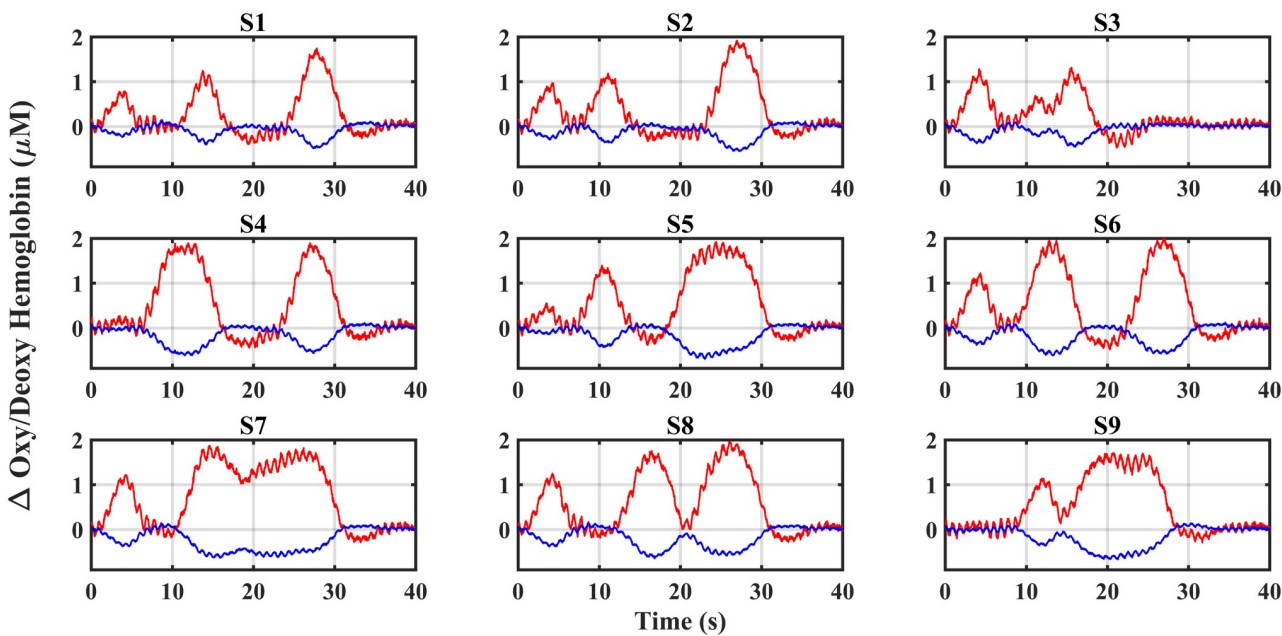

**Fig 5. Changes in synthetic oxyhemoglobin ($\Delta(HbO_2)$) and deoxyhemoglobin ($\Delta(Hb)$).**

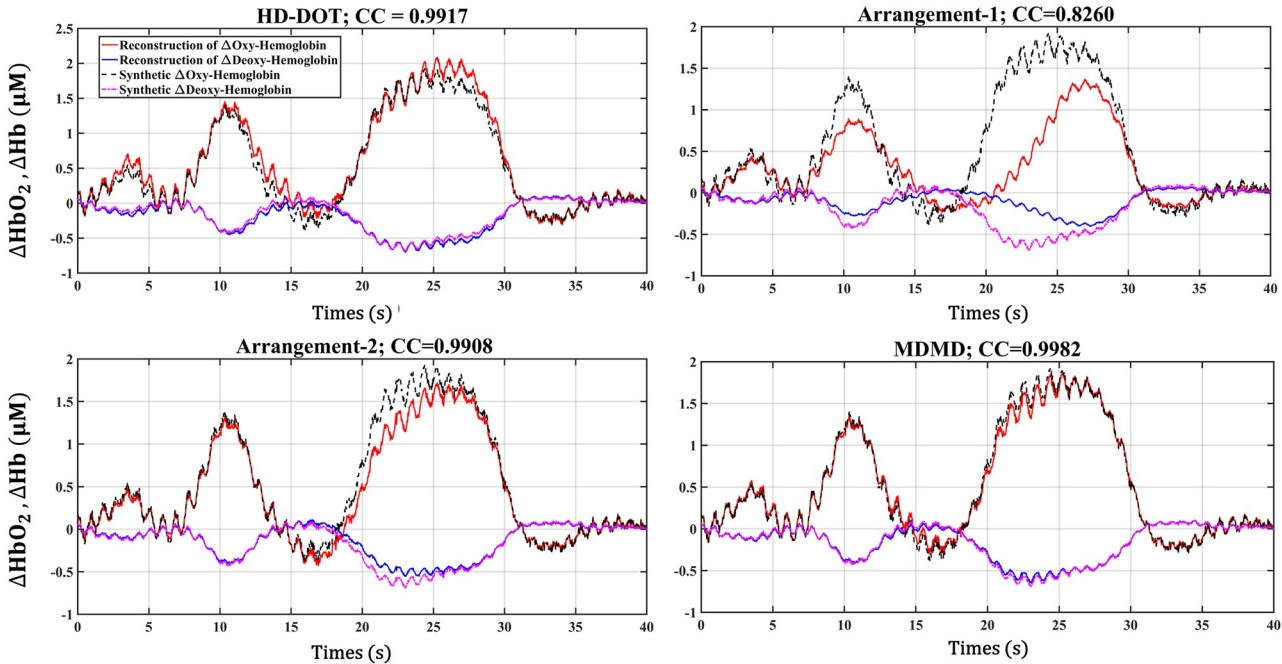

**Fig 6. Visual comparison between reconstructed and synthetic hemodynamics for S5.**

inverse algorithm and the SD arrangement can be explored. It can be noted that the nearby hemodynamic activity acts as a systematic noise. Therefore, the hemodynamic recovery of the S5 region will be more difficult because it is surrounded by eight hemodynamic noises. Thus, for hemodynamic retrieval of this area, the number and angle of observations must be much higher than the number of hemodynamic sources under the inspection field.

## Inverse procedure: Hemodynamic reconstruction

The correlation rate was used to investigate the similarity between the two synthetic and obtained hemodynamic. The accuracy of reconstruction not only depends on the arrangement and number of the SD, but it also relies on the solution of the energy regularization. The optimum value for the energy regularization parameter ($\epsilon$) was achieved by sweeping this parameter from $10^{-8}$ to $10^{-3}$ and minimizing the cost function; correlation coefficient (CC) for four different SD topology. By using Eqs (13) and (14), and reconstructed $\Delta\mu_a(\vec{r}, \lambda, T_s)$ in two wavelengths, the calculated Oxy-Deoxy hemoglobin along with synthetic data are compared in the S5 region in Fig 6 (in the existence of SD arrangement of HD-DOT, Arrangement-1, Arrangement-2, and MDMD).

Fig 6 shows that the topologies of Arrangement-2, HD-DOT, and MDMD have been able to extract the hemodynamics of the S5 region with greater accuracy, but the Arrangement-1 has inferior performance compared to other configurations.

The trend of CC($\epsilon$) within Fig 7 represents that change in SD topology leads to accurate hemodynamic reconstruction. It also indicates that the MDMD has superior performance compared to other arrangements. The production of each SD arrangement also analyzed in the rebuilding of all dynamic perturbations of S1-S9 using the value of CC. The result of this comparison, as shown in Fig 8, reveals that almost all the combinations have acceptable performance except for the Arrangement-1, which has poor performance in hemodynamic

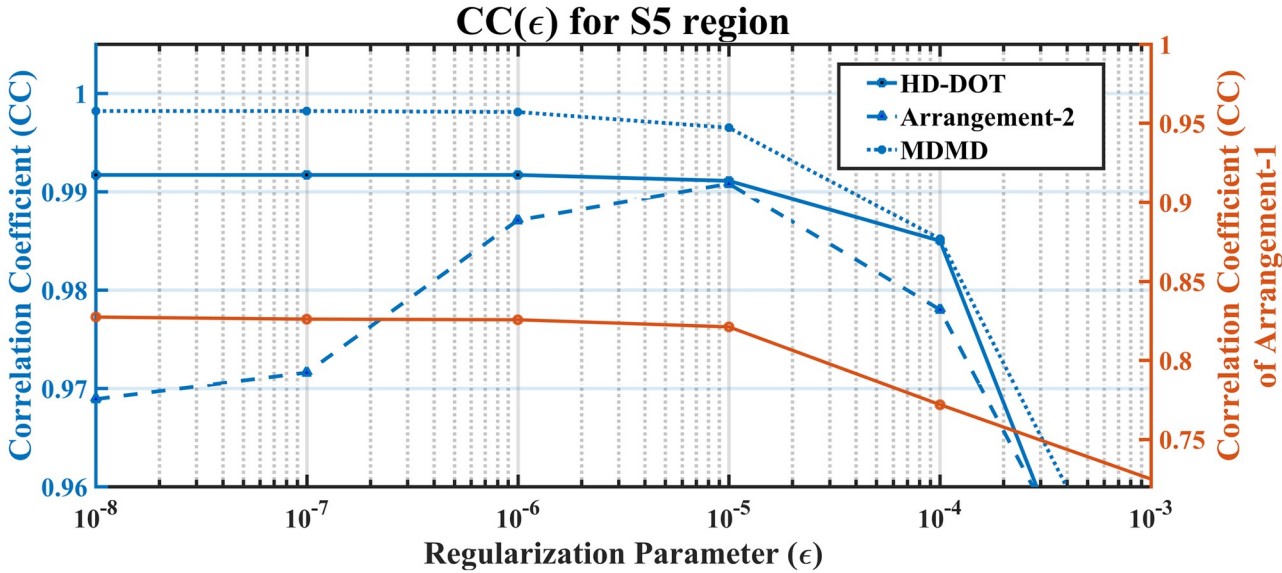

**Fig 7. Depicts the performance assessment of different SD arrangements in hemodynamic extraction of the S5 region concerning the regularization parameter.**

extraction of the S5 region. This figure also confirms that the MDMD operates properly in extracting all the hemodynamic sources because it has many unique elements in the Jacobian matrix compared to other topologies (Fig 9).

The performance of all the topologies studied in this work is summarized in Table 1 in terms of the topology of the arrangement, the number of SDs, channels, distances, directions, as well as the unique elements of the Jacobian matrix. This table notes that as much as the

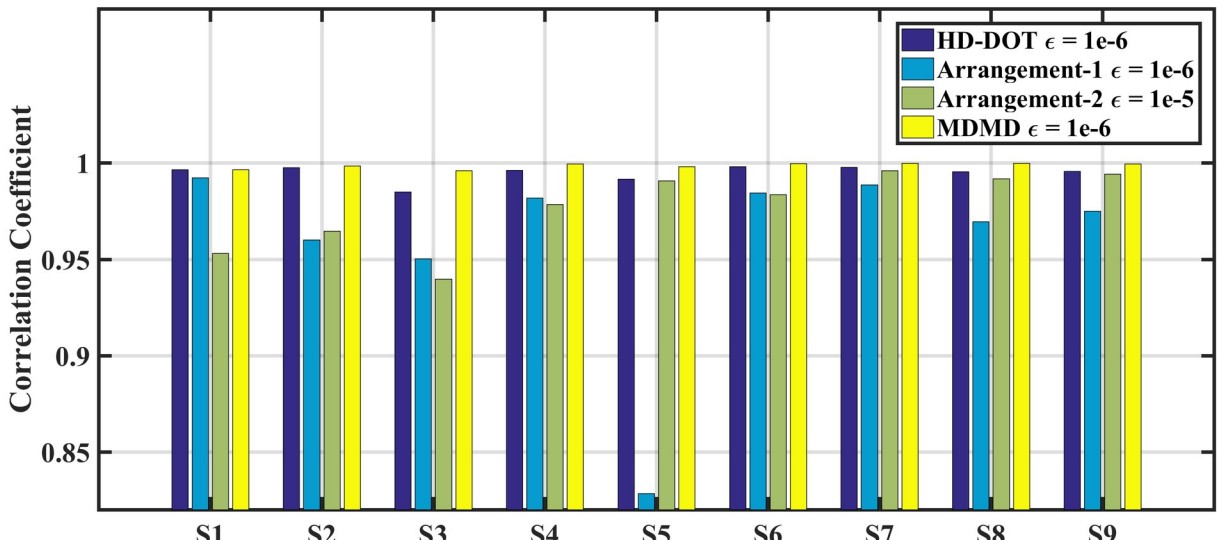

**Fig 8. Indicates the similarity of all reconstructed and synthetic hemodynamic in S1-S9 region for HD-DOT, Arrangement-1, Arrangement-2 and MDMD.**

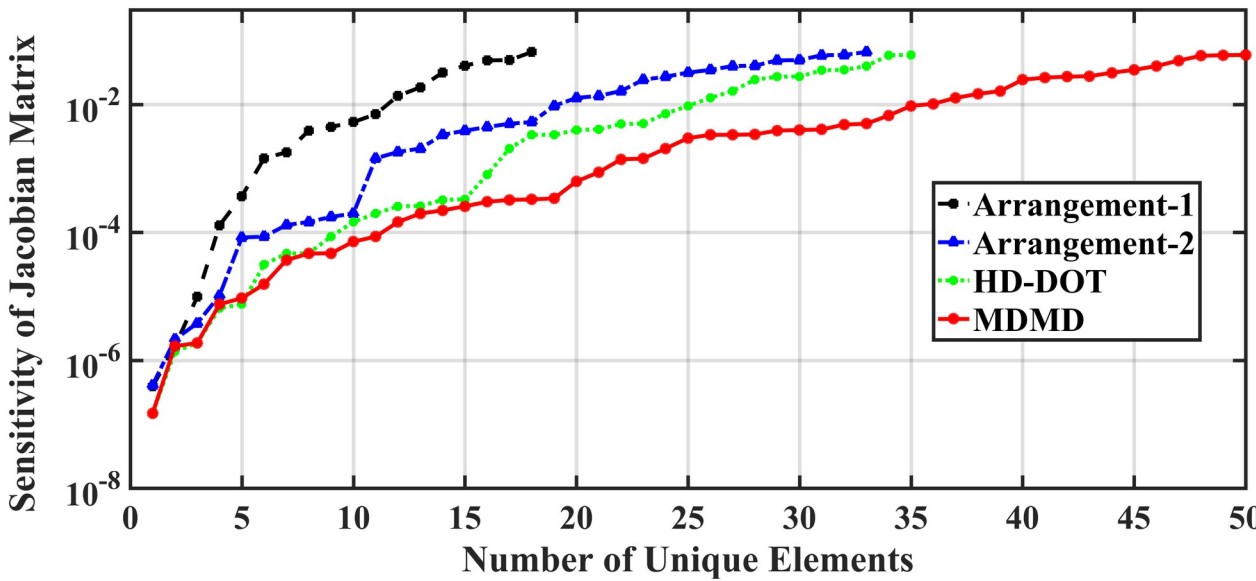

**Fig 9. Represents the number of unique elements in the Jacobian matrix regarding the SD topology.**

arrangement of the SDs creates various distance and direction between the overlapped channels, the better the hemodynamic extraction will be observed.

A singular value analysis of the Jacobian matrix associated with introduced SD arrangements is used as a benchmark [30]. Besides unique elements of the Jacobian matrix, the singular value analysis of the different methods in Fig 10 indicates that both shape of the singular value spectra and the magnitude for MDMD arrangement is higher than other SD combinations.

It is worth noting that, if the depth information is not necessary, and the objective is to achieve topography, Arrangement-2 can be replaced instead of MDMD and HD-DOT because it has fewer SDs (reduces the complexity of the device) and has acceptable performance compared to these arrangements.

## Discussion

In this paper, based on a developed simulation setup, the performance of SD arrangement and their quantity alongside inverse problem on hemodynamic reconstruction has been investigated. The simulation approach consists of a forward model, synthetic fNIRS data generation, Inverse problem, and SD arrangement.

**Table 1. Summarized the details and the performance of all SD topologies studied in this investigation.**

| SD Topology | Number of Distance | Number of Direction | Number of SD | Number of Channels | Number of Unique Elements in Jacobian Matrix | Total Correlation Coefficient |
|---|---|---|---|---|---|---|
| Arrangement-1 | 2 | 6 | 9 | 12 | 18 | 0.9586 |
| Arrangement-2 | 4 | 3 | 9 | 16 | 33 | 0.9764 |
| HD-DOT [10] | 4 | 9 | 13 | 36 | 35 | 0.9949 |
| MDMD | 5 | 13 | 13 | 36 | 50 | 0.9986 |

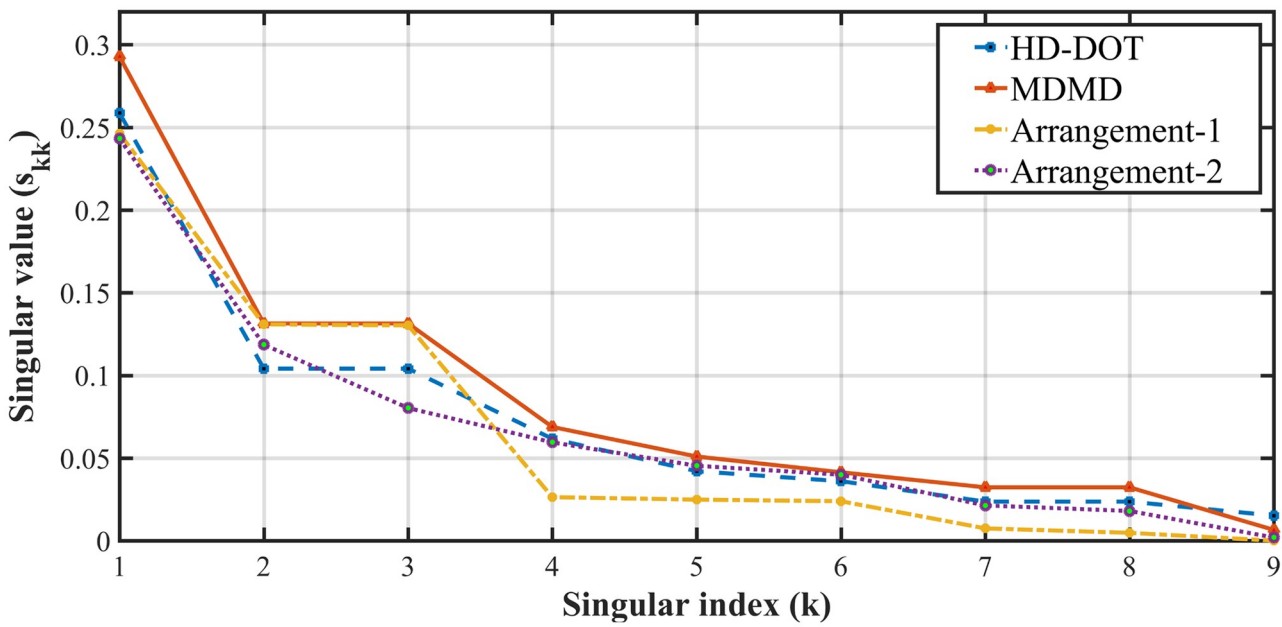

**Fig 10. Singular value spectra for Jacobian matrix of HD-DOT, MDMD, Arrangement-1 and 2.**

The forward model is an analytical method that is implemented by the solution of the pDE in slab medium. Analytical methods have been developed earlier to study light emission inside the simple geometry such as slab medium [13,31–36]. Numerical methods also are used in complex brain models to study light diffusion in tissues [14,37]. In spite of simplicity and approximation, analytical methods take less time calculation compared to statistical approaches, especially when it comes to investigating the effect of several fNIRS channels on depth sensitivity.

The spatial probability pattern of photons penetrating tissue at the source position, scattering within the tissue, and being exposed at a particular detector spot, determines the spatial sensitivity profile for the SD pairs [14]. To verify the forward model, the spatial sensitivity profile is compared with the results of the Monte Carlo method on Colin 27 brain template. The depth sensitivity for analytical pDE inside slab geometry is defined as follow:

$$Depth\ Sensitivity\ (z, x_{SD}) = \frac{\sum_{x=0}^{x_{SD}} \delta R^a(x, y = 0, z)}{((x_{SD})/1mm) \times R^0(x_{SD})} \tag{15}$$

Where $x_{SD}$ represents the distance between source and detector.

The depth sensitivity of analytical pDE is compared to the regression of Monte Carlo (MC) on Colin 27 brain template in Fig 11, and the mismatch between these analytical and numerical methods are illustrated in this figure for given SD separations. Toward SD separation around 30mm-50mm, the mismatch is less than 60% for penetration depth from 1mm to 28mm. In 15mm depth, the mismatch is less than 20% for SD separation of 30mm-50mm.

The outcomes of the comparison indicate that the analytical approach is not too close to the results of Monte Carlo methods. It is better to emphasize that there is no analytical solution to light- tissue interaction inside complex geometries like the brain. Besides that, due to the high computation time of statistical models are not a desirable candidate to be a forward model in this simulation setup. Although Monte Carlo methods are accurate in estimation of depth sensitivity, they are not well suited to be used in the forward model. Whenever multiple dynamic

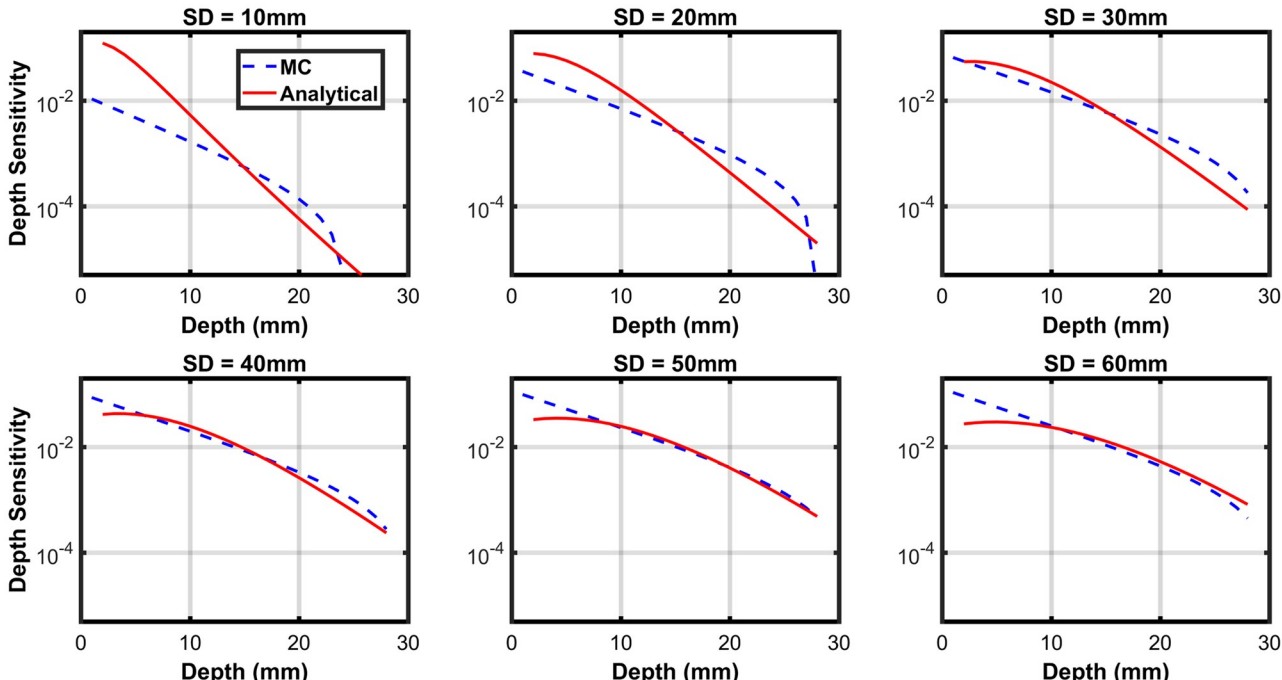

**Fig 11. Represents the depth sensitivity of the analytical pDE in slab medium and depth sensitivity of the regression of MC on Colin 27 geometry for SD separation of 10mm-60mm.**

perturbations exist in geometry, then numerical methods should be run for each sampling time.

Consequently, the computation time will grow significantly by applying statistical approaches. It is suggested to try the solution of analytical techniques on multilayered geometries as a forward model inside the simulation setup. Another alternative to the computation of the forward model and Jacobian matrices is to use a finite element method (FEM) [27,36]. Recently, both NIRSFAST and Neuro-DOT software have been developed to the solution of the forward model based on FEM estimation [38–41].

The results of FEM data when it is applied to Diffusion Equation will be close to reality compared to the analytical techniques, but the computation time will grow considerably [15]. On the other hand, adding dynamics to the perturbations inside the meshed environment will increase the simulation time significantly. Analytical simulation of 36 channels, including a sampling rate of 30 samples per second, takes less than 10 minutes. For the same sampling rate and channels, the FEM simulation will take almost 43–48 hours, depending on system speed. The geometry and mesh of the medium for simulating the FEM are represented in Fig 12(a) and 12(b), respectively.

According to the simulation approach presented in Fig 1, White Gaussian Noise (WGN) is added to each channel $(\Delta R_j^{pert}(\rho_j, T_s)/ R_i^0(\rho_j))$ before the reconstruction of the simulated data to avoid inverse-crime. WGN indicates the instrumentational noise, which depends on the Signal to Noise Ratio (SNR) of the signal acquisition device. The effect of 47 dB SNR on the performance of different SD arrangements in the hemodynamic recovery of region S5 has been investigated. Considering the simulation of Fig 13 for Arrangement-1 and 2, there is a significant change in magnitude and shape of CC versus regularization parameter. While little difference with the noiseless condition for MDMD and HD-DOT is observed, it can be

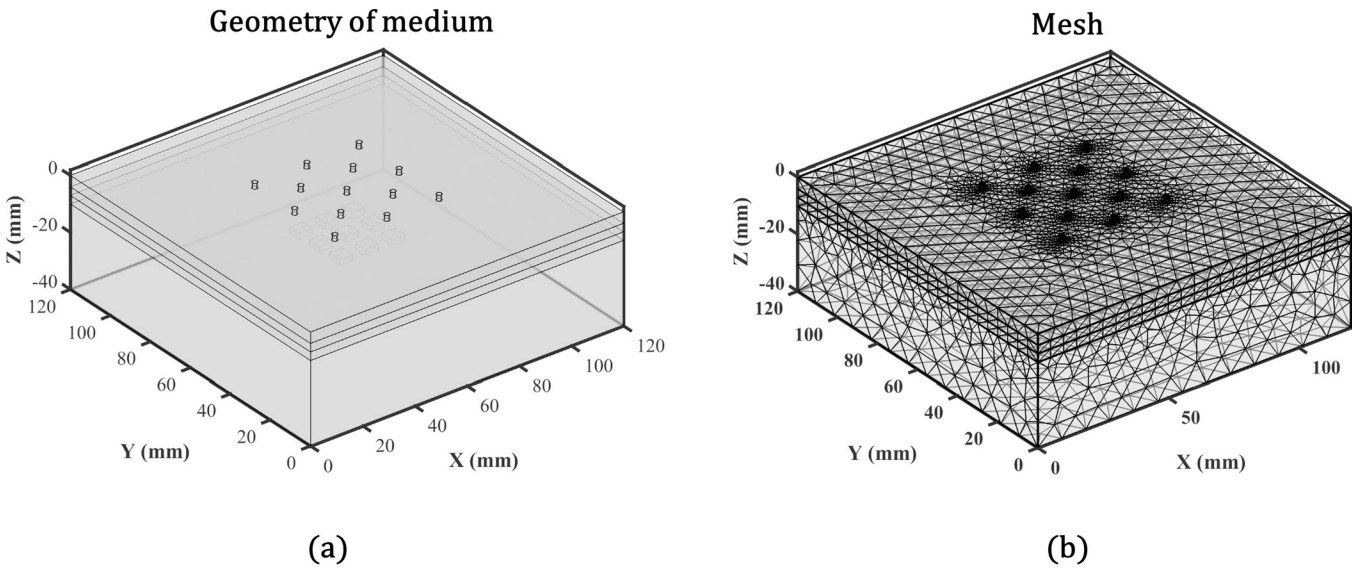

**Fig 12.** (a) Illustrates the geometry of the Slab medium which is 120mm × 120mm × 40mm. (b) represents the mesh of the medium.

concluded that 47 dB of SNR has no significant effect on the performance of these two methods. The SNR has been swept from 32 dB (worst case condition) up to 52 dB (for given $\epsilon = 10^{-5}$) to compare the performance of SD arrangements on hemodynamic reconstruction. The simulation of Fig 14(a) represents that for any SNR, MDMD performs better than other competitors. Concerning Fig 14(b) in the worst-case condition, the optimal point for the regularization parameter has been shifted. Even with the highest noise through determining the appropriate regularization parameter, MDMD still works better than HD-DOT. For SNR = 32

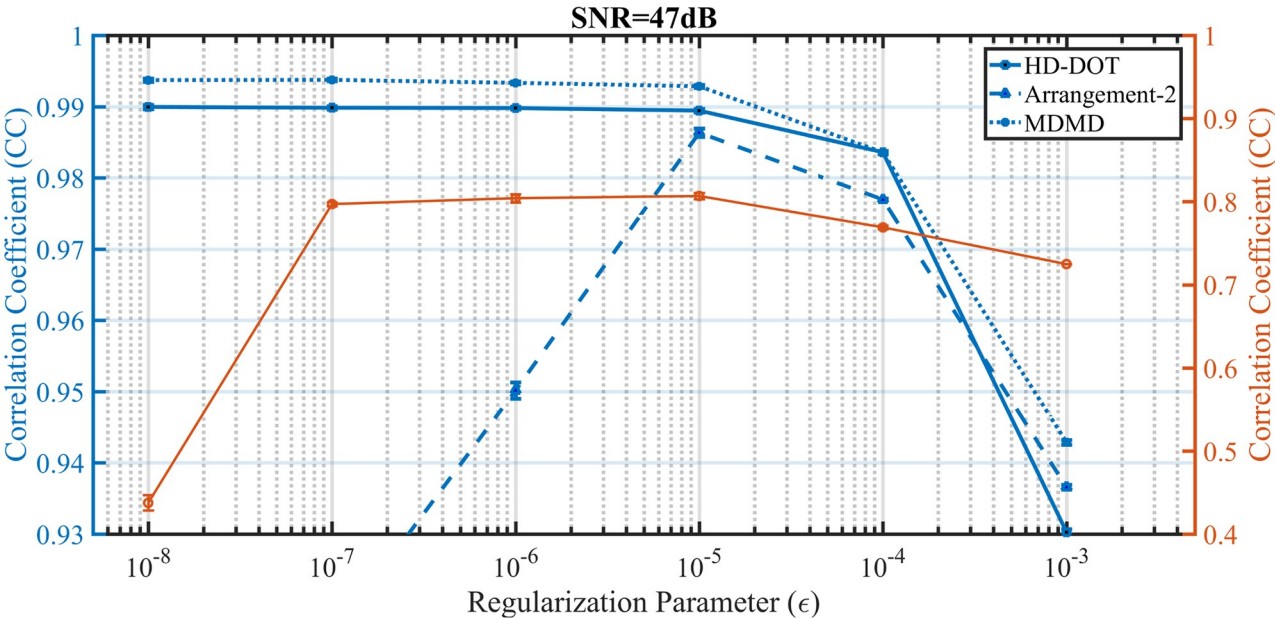

**Fig 13. Demonstrates the effect of noise on performance assessment of different SD arrangements in hemodynamic extraction of the S5 region concerning the regularization parameter.**

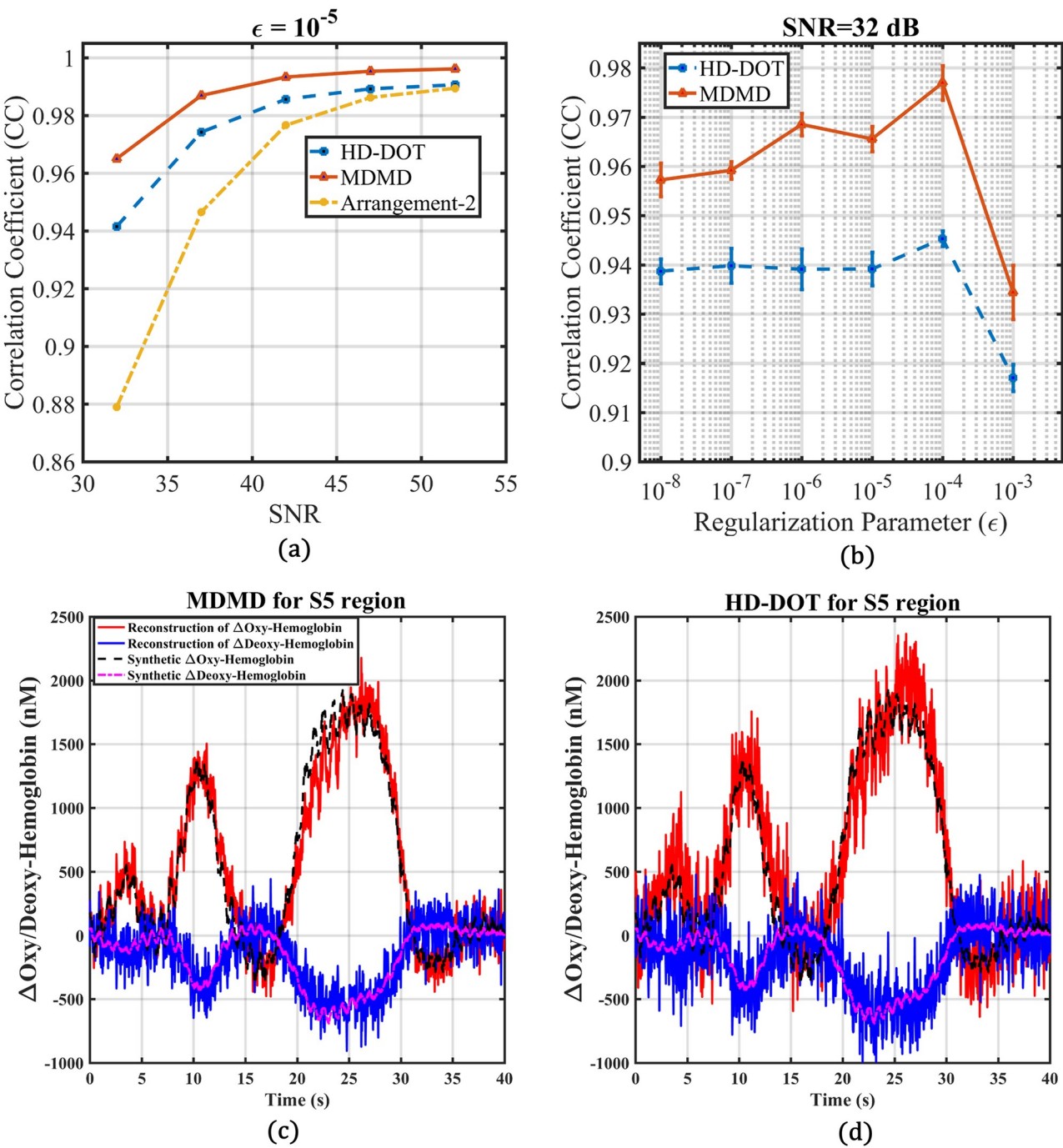

**Fig 14.** (a) Represents the ability of hemodynamic extraction from the S5-region concerning several SNRs. (b) illustrates that in the worst case condition (SNR = 32 dB), the optimum point for $\epsilon$ has changed from $10^{-5}$ to $10^{-4}$. (c) and (d) Represents reconstructed and synthetic $\Delta HbO_2$ and $\Delta Hb$ for MDMD and HD-DOT respectively.

dB, the reconstructed and synthetic hemodynamic of region S5 are presented in Fig 14(c) and 14(d) for both MDMD and HD-DOT, respectively.

The forward model has been generated with well-defined optical properties using the Jacobian matrix, and the inversion has been performed using the same matrix. Previously, to avoid

inverse-crime, WGN was added to each channel. One can also bring the forward model closer to the more realistic model by adding uncertainty to all elements of the Jacobian matrix. For this purpose, Jacobian matrix elements are multiplied by the Gaussian random coefficient. Therefore the forward model is changed as follows:

$$\frac{\Delta R_j^{pert}(T_s)}{R_i^0(\rho_j)} = Rnd_{j\times i}. \, J(\mu_a(\vec{r}))_{j\times i} \, \Delta\mu_a(\vec{r}_i, T_s) \tag{16}$$

The Rnd matrix carries random coefficients with Gaussian distribution, the performance of the two MDMD and HD-DOT methods are close together, the effect of the changes on the forward model will only be investigated on the performance of these two methods.

Random coefficients with two different distributions are applied to the forward model. The mean of both data set is one, and the variance ($\delta$) of the first and second random coefficients are 0.07 and 0.2, respectively (Fig 15(a) and 15(c)). Beside ten simulation runs for 0.07 variance, the distribution of random coefficients is plotted in Fig 15(a). The performance of both the MDMD and HD-DOT methods in hemodynamic extraction of the S5 region is compared in Fig 15(b). Similarly, for the variance of 0.2, the above comparison is repeated 18 times. In this case, the distribution of random data for this variance is shown in Fig 15(c) and the results of analyzing of similarity are illustrated in Fig 15(d).

If the variance of random data in the Rnd matrix is 0.07, according to the results of Fig 15 (b), the performance of MDMD is better than HD-DOT. Even with a variance of 0.2, MDMD still delivers better results. It should be noted that there is no similarity between the data extracted from the S5 region and the synthetic data under the variance of 0.2 in some states. If the correlation is less than 0.8, there will be no similarity between the reconstructed and synthetic signals.

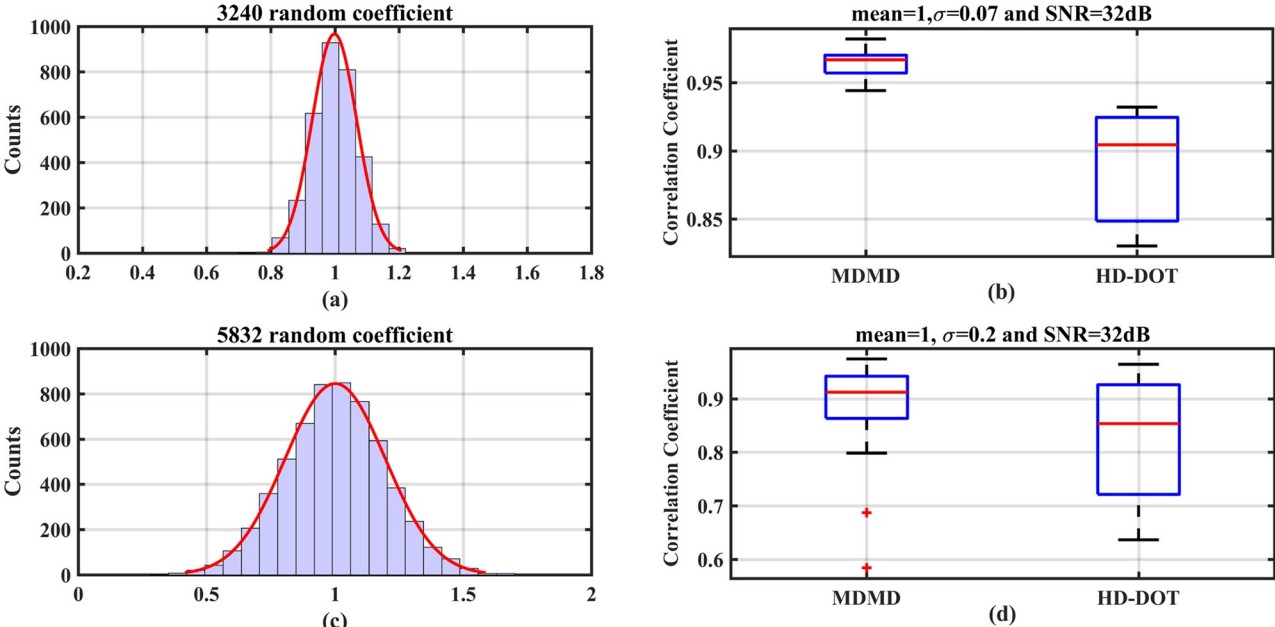

**Fig 15.** (a) Represents Gaussian distribution (mean = 1 and $\delta$ = 0.07) of Rnd matrix elements. (b) Boxplot regarding random coefficients. (c) Represents Gaussian distribution (mean = 1 and $\delta$ = 0.2) of Rnd matrix elements. (d) Boxplot regarding random coefficients.

Although adding uncertainties to the Jacobian matrix elements in the forward model can reduce the gap between the results of this work and the real imaging applications. However, given the limitations of the analytical model used in this comparison, it is not yet possible to claim that MDMD will have better results than other SD configurations in imaging applications.

This proposed simulation setup is expandable for many numbers of perturbation inside the medium and can be used for performance assessment of HD-DOT. A. Eggebrecht and colleagues in 2014 have used HD-DOT for mapping brain function [10], the proposed model can be used for evaluation of the SD separation and arrangement of SD on performance of HD-DOT.

The outcome of this model can be an instrumentation probe with a specific arrangement of SD array for monitoring stimulation induced hemodynamic. Among noninvasive stimulation approaches such as transcranial direct current stimulation and transcranial magnetic stimulation, low-intensity ultrasound stimulation has the spatial resolution in the order of several millimeters [42]. To control the amount of stimulation and study the effect of stimulation on the brain, a simultaneous recording of the hemodynamic activity of the brain is necessary [43]. Therefore, a non-invasive method is required for recording stimulation-induced hemodynamic with a spatial and temporal resolution equivalent to the stimulation approach [44].

Summarily, this simulation setup can be employed for performance assessment of high or low-density DOT, monitoring of stimulation-induced hemodynamic, and SD array design. Finally, the proposed simulation approach, with declared assumptions and simplifications, can be used by researchers who want to arrange an array of sources and detectors for optical topography.

## Conclusion

In this paper, an innovative simulation setup proposed for the performance assessment of a variety of sources and detectors toward the rebuilding of cerebral hemodynamics. MDMD arrangement involves more unique elements in the Jacobian matrix and will be able to reconstruct the neural activity accurately. Meanwhile, the performance of several provisions of SD on the reconstruction of brain function is studied. The result of simulation indicates that raising the number of multi-distance and multi-directional overlapped channels increase the accuracy of brain hemodynamic reconstruction. Also, the simulation setup can be employed for performance assessment of high or low-density DOT, monitoring of stimulation-induced hemodynamic, and SD array design. We believe that modeling and simulation of different SD arrays on hemodynamic extraction optimizes the number of SDs required for accurate spatial imaging. Consequently reduces the additional cost and complexity of device fabrication. Based on the modeling approach and simulation results, the achievements of the MDMD looks more beneficial than other methods. But still, there would be a gap between the outcomes of this study and real imaging applications. It is suggested to use the proposed simulation approach with a modified forward model according to the following suggestions. The brain model in this work is one-layer, while the multi-layered medium can be considered to get closer to the real results. Also, we must find the interaction between the perturbations in the solution of DE equations (whenever the perturbations are not small regarding the baseline optical properties), while Born approximation is used in this work. In this study the diffuse reflectance modulated due to change in absorption coefficient. The changes in the optical scattering coefficient, along with the absorption coefficient, should be taken into account for more accurate simulation of diffuse reflectance.

## Supporting information

**S1 Appendix.**
(DOCX)

**S1 Data.**
(RAR)

## Acknowledgments

The authors would like to thank Dr. Alessandro Torricelli, professor of the department of physic at the University of Politecnico di Milano for his valuable suggestions, many helpful hints, and scientific supports on developing this manuscript.

## Author Contributions

**Conceptualization:** Hadi Borjkhani, Seyed Kamaledin Setarehdan.

**Data curation:** Hadi Borjkhani, Seyed Kamaledin Setarehdan.

**Formal analysis:** Hadi Borjkhani, Seyed Kamaledin Setarehdan.

**Investigation:** Hadi Borjkhani, Seyed Kamaledin Setarehdan.

**Methodology:** Hadi Borjkhani, Seyed Kamaledin Setarehdan.

**Project administration:** Seyed Kamaledin Setarehdan.

**Resources:** Hadi Borjkhani, Seyed Kamaledin Setarehdan.

**Software:** Hadi Borjkhani, Seyed Kamaledin Setarehdan.

**Supervision:** Seyed Kamaledin Setarehdan.

**Validation:** Hadi Borjkhani, Seyed Kamaledin Setarehdan.

**Visualization:** Hadi Borjkhani, Seyed Kamaledin Setarehdan.

**Writing – original draft:** Hadi Borjkhani, Seyed Kamaledin Setarehdan.

**Writing – review & editing:** Hadi Borjkhani, Seyed Kamaledin Setarehdan.

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
