## [Decision Letter · Decision Letter 0]

20 Dec 2019

PONE-D-19-31493

Performance Assessment of High-Density Diffuse Optical Tomography Regarding Source-Detector Array Topology

PLOS ONE

Dear Mr. Setarehdan,

Thank you for submitting your manuscript to PLOS ONE. After careful consideration, we feel that it has merit but does not fully meet PLOS ONE’s publication criteria as it currently stands. Therefore, we invite you to submit a revised version of the manuscript that addresses the points raised during the review process.

We would appreciate receiving your revised manuscript by Feb 02 2020 11:59PM. To enhance the reproducibility of your results, we recommend that if applicable you deposit your laboratory protocols in protocols.io, where a protocol can be assigned its own identifier (DOI) such that it can be cited independently in the future. For instructions see: http://journals.plos.org/plosone/s/submission-guidelines#loc-laboratory-protocols

We look forward to receiving your revised manuscript.

Kind regards,

Alberto Dalla Mora, Ph.D.

Academic Editor

PLOS ONE

Journal Requirements:

1.Please ensure that your manuscript meets PLOS ONE's style requirements, including those for file naming. The PLOS ONE style templates can be found at http://www.plosone.org/attachments/PLOSOne_formatting_sample_main_body.pdf and http://www.plosone.org/attachments/PLOSOne_formatting_sample_title_authors_affiliations.pdf

Reviewers' comments:

Reviewer's Responses to Questions

**Comments to the Author**

1. Is the manuscript technically sound, and do the data support the conclusions?

Reviewer #1: Yes

Reviewer #2: Yes

Reviewer #3: Yes

Reviewer #4: No

2. Has the statistical analysis been performed appropriately and rigorously? 

Reviewer #1: N/A

Reviewer #2: No

Reviewer #3: Yes

Reviewer #4: N/A

3. Have the authors made all data underlying the findings in their manuscript fully available?

Reviewer #1: Yes

Reviewer #2: No

Reviewer #3: Yes

Reviewer #4: Yes

4. Is the manuscript presented in an intelligible fashion and written in standard English?

Reviewer #1: Yes

Reviewer #2: Yes

Reviewer #3: Yes

Reviewer #4: Yes

5. Review Comments to the Author

Reviewer #1: The following paper addresses the effect on optical tomography of the source-detector array configurations. The work is done in the Continuous Wave domain and is based on simulated data. Experimental results have not been presented. The forward solver used is based on an analytical solution of the diffusion equation. It has been considered a perturbative solution for the slab in reflectance configuration where an absorption inclusion is inserted inside. The Born approximation is used to obtain the solution of the perturbed reflectance. The results presented show that the multi-distance multi-directional (MDMD) arrangement of sources and detectors produces more unique elements in the Jacobian matrix and consequently the related inverse problem can better retrieve the brain activity of diffuse optical tomography data compared to other arrangements of sources and detectors. The results are corroborated by several retrievals obtained making use of the mentioned forward model applied on synthetic data generated with the same model on which noise is applied.

These results appear scientifically sound and I suggest the publication on PLOS ONE provided the following remarks are addressed.

The results presented in this paper represent a proof of principle since they are obtained by inverting synthetic data generated by using the same forward solver employed in the inversion procedure and with an added noise accordingly to Eq. (7). Thus, they can be considered a first step of study. At this stage arise a question that deserve at least some comments and explanations. Do the improvements obtained by using MDMD hold in all generality when applied to real cases as real experiments on biological tissues, for instance on brain? Compared to the results presented in this paper in real imaging applications the forward solver used in the inversion procedure can show some deficiencies since the diffusion equation and the Born approximation have limitations and the background medium addressed is homogeneous differently from real media. The gap between the case here addressed and the real applications can have some consequences in the conclusions here formulated? I can understand that the MDMD arrangement can be anyway advantageous for the purpose of imaging applications, however some comments should be spent on this fact. It is true that the authors present a comparison of the depth sensitivity of their analytical model with the Monte Carlo data on Colin 27 brain template. However, the results shown in Fig. 10 and 11 emphasizes difference that, although smaller for larger depth, are always present between analytical model and Monte Carlo regression. The authors simply conclude that Monte Carlo methods although accurate in the estimation of depth sensitivity are not well suited to be used in the forward model due to their long computation time. Is it possible to conclude that the differences observed are not decisive in the inversion procedure?

Minor points:

1) At line 60 the single Ref. 15 is not enough, I would also add Ref. 16.

2) At line 62 it is written “… simple slab boundary condition in semi-infinite geometry are utilized [16]”. The sentence is confusing in the sense that is confused the meaning of geometry and boundary condition. Here the term boundary condition appears misleading. To me would make sense to rephrase as “… the simple geometry, that can approximate a semi-infinite medium for thick slabs, is utilized [16]”

3) At line 96 it is written “The analytical solution for perturbative DE has been obtained in the boundary condition of Fig. 3.” So far, I understand it should be “The analytical solution for perturbative DE has been obtained for the geometry of Fig. 3.” Boundary condition and geometry have a different meaning in this context. The actual boundary condition used to solve DE for the forward solver used in this paper is the extrapolated boundary condition that is not a geometry but a condition to make an energy balance at the external interface of the medium.

4) At line 102 the symbol S (The thickness of the slab) is not defined. NB that S is also the symbol used for the different regions S1-S9 (See Fig. 3).

5) At line 104 is written “Consequently, the volume of the inhomogeneity (inclusion) is regarded to be small concerning baseline optical properties of the homogeneous medium [16].” It should be “Consequently, the volume of the inhomogeneity (inclusion) is regarded to be small concerning baseline optical properties of the homogeneous medium [16].”

6) At line 112 it is written “The final expression of Rpert(\\rho) for each channel derived based on Born approximation [21].” It looks like the sentence miss the final part. The sentence recalls the Born approximation; however, Ref. 21 is mainly related to higher order perturbation theory. Do the authors mean that in Ref. 21 the results for the Born approximation are also presented?

7) At line 136 the acronym HRF is used without definition. I understand that implicitly means Hemodynamic Response Function, however why it should be omitted this definition?

8) At line 272, Eq. (15), index of the sum in Eq. (15) is a real number, while has not been used an index numbering the number of Source-Detector pairs?

9) At line 292, it is mentioned that the results of FEM will be closer to reality than analytical techniques. Maybe could be worth to note that to some extent also FEM data when FEM is applied to the Diffusion Equation, since the intrinsic approximations of this theory affects the FEM data.

10) The actual title of Ref. 16 is: “Light Propagation through Biological Tissue and other Diffusive Media: Theory, Solutions and Software” 2009.

11) In the first row of Fig. 4 is missed the info on the y axis (Z (mm)?).

12) In the second row of Fig. 6 is missed the info on the x axis (Time s?).

Reviewer #2: Author report on a simulation platform for diffuse optical tomography (DOT) adapted for reconstruction of hemodynamic responses. The platform is based on analytical solutions, under the Born approximation, of the diffuse equation for a semi-infinite slab. The work is technically sound and, together with the claimed availability of software and data, will be useful for setting up DOT systems.

At my opinion there are the following points to address and clarify:

Reconstruction: what I don’t understand is whether the reconstruction is forced at the depth of 15 mm or it is performed in the whole volume. In the first case, the point has to be emphasized and better specified in the text and, at my opinion, the method can’t be properly called “tomography”. In the second case, as well, it has to be emphasized in the text and, what I expect, is a figure representing slices in the volume at a defined time Ts to see the reconstructed depth that, in DOT, is typically underestimated.

Noise and Inverse-crime: As far as I’ve understood, the only noise added is on the optical properties of the S5 and nearby voxels. This means that the forward model has been generated with well-defined optical properties using the Jacobian matrix, and the inversion has been performed using the same matrix on unnoisy data ΔR. This is typically called an “inverse-crime”. I suggest to add noise (Gaussian or Poisson) to the simulated data to avoid this problem.

Minors:

the acronym HRF is not specified in the text.

Figure 1 is reported with a very poor resolution, writings are too small.

Reviewer #3: The authors present a simulation setup for the performance assessment of different source-detector configurations in high density diffuse optical tomography. The information content in the paper is well organised. While the work presented is interesting and very useful research, a few concerns are raised below, which are to be addressed prior to any publication.

Major concerns:

1) One of the important aspects of the presented work is the significant reduction in computational time due to the use of analytical forward model, when compared to finite element and Monte Carlo methods. Therefore, it is crucial to provide the readers with the comparison of individual computational times for a sample forward model.

2) From Figure 9, and Table-1, the authors want to show that higher the number of unique elements in Jacobian matrix, better will be the recovery. This can be misleading. While it is logical that a Jacobian with more independent information can give a better recovery, the correct way to observe this is to compare the normalised singular values of different Jacobian matrices corresponding to the SD arrangements. I would recommend looking into: Optics Letters Vol. 26, Issue 10, pp. 701-703 (2001).

3) Figure 10 and 11 have same information, therefore figure 10 can be avoided and two additional sub-plots for SD separation 10 and 20mm can be included in Figure 11.

Minor concerns:

1) Add reference for lines 150-152.

2) Use one abbreviation for oxy hemoglobin (either O2Hb or HbO2), and for deoxy hemoglobin (either HHb or Hb) throughout the manuscript.

3) Important future directions such as the use of multi-layered medium, fit better in the conclusion section rather than discussions. Re-edit the conclusion section to incorporate this information seamlessly.

Reviewer #4: In the manuscript, ‘Performance Assessment of High-Density Diffuse Optical Tomography Regarding Source-Detector Array Topology,’ the authors present a simulation study comparing various source-detector separation distances. While the methods are sound, the choice of methods (Born approximation, limited number of source-detector pairs and therefore a small field of view, analytical model instead of anatomy-based FEM using either diffusion or MonteCarlo, the single layer of optical properties) leads to the results and discussion providing limited information to the community for further advances in simulation, system design, or empirical considerations. As such, the authors are encouraged to add complexity to their modeling procedures or better motivate their choices to place their work in context with the current status of the field.

6. PLOS authors have the option to publish the peer review history of their article (what does this mean?). If published, this will include your full peer review and any attached files.

Reviewer #1: No

Reviewer #2: No

Reviewer #3: No

Reviewer #4: No

---

## [Author Response · Author response to Decision Letter 0]

31 Jan 2020

Dear Professor A. Dalla Mora

We would like to thank you and the esteemed reviewers for providing us with insightful comments, which helped us to revise and improve the quality of the manuscript. We have addressed all the comments, as shown in the revised manuscript. We believe that the contents and the clarity of our paper are much improved in the revised version. Below are point-by-point responses for the four reviewer’s comments. Finally, all the responses in the “Response to Reviewers” have been highlighted in blue. 

Editor Comments

Point1: Please ensure that your manuscript meets PLOS ONE's style requirements, including those for file naming. The PLOS ONE style templates can be found at http://www.plosone.org/attachments/PLOSOne_formatting_sample_main_body.pdf and http://www.plosone.org/attachments/PLOSOne_formatting_sample_title_authors_affiliations.pdf

Response: Thanks. The revised manuscript has been edited to meet PLOS ONE’s standards.

Point2: We note that you have stated that you will provide repository information for your data at acceptance. Should your manuscript be accepted for publication, we will hold it until you provide the relevant accession numbers or DOIs necessary to access your data. If you wish to make changes to your Data Availability statement, please describe these changes in your cover letter and we will update your Data Availability statement to reflect the information you provide.

Response: The simulation codes would be available according to PLOS ONE’s policies.

Authors hint to Editor and all Reviewers:

In this response, some simulation results are not included in the revised manuscript. They only are added to make the responses more clear. Those figures from the revised manuscript are highlighted inside the green box. To avoid possible confusion, please consider this hint before reading the answers.

Reviewer #1:

Comments

The following paper addresses the effect on optical tomography of the source-detector array configurations. The work is done in the Continuous Wave domain and is based on simulated data. Experimental results have not been presented. The forward solver used is based on an analytical solution of the diffusion equation. It has been considered a perturbative solution for the slab in reflectance configuration where an absorption inclusion is inserted inside. The Born approximation is used to obtain the solution of the perturbed reflectance. The results presented show that the multi-distance multi-directional (MDMD) arrangement of sources and detectors produces more unique elements in the Jacobian matrix, and consequently, the related inverse problem can better retrieve the brain activity of diffuse optical tomography data compared to other arrangements of sources and detectors. The results are corroborated by several retrievals obtained making use of the mentioned forward model applied to synthetic data generated with the same model on which noise is applied.

These results appear scientifically sound and I suggest the publication on PLOS ONE provided the following remarks are addressed.

Response: We appreciate the positive feedback of the esteemed reviewer.

 Major Concerns:

Point 1: The results presented in this paper represent a proof of principle since they are obtained by inverting synthetic data generated by using the same forward solver employed in the inversion procedure and with an added noise accordingly to Eq. (7). Thus, they can be considered a first step of study. At this stage arise a question that deserve at least some comments and explanations. Do the improvements obtained by using MDMD hold in all generality when applied to real cases as real experiments on biological tissues, for instance on brain?

Response: Thanks to the reviewer for raising these points. In this manuscript we have tried to develop a simulation model to understand more about the imaging process and to see the strengths and weaknesses of different blocks such as forward model, SD arrangement and inverse problem.

As you mentioned, this manuscript is the first step of the study. There should be a gap between the results of this investigation with the experiment. The only problem is that in real cases, we do not have access to the inside brain; in another word, access to exact hemodynamic change is not possible. It may need another modality such as ECoG, which is invasive. If we apply different arrangements to tissue-like phantom (as a forward model), there may be a chance to produce synthetic hemodynamic, but it is still challenging. The challenge is how to insert dynamic perturbations inside the phantom. In the Analytical solution, we can add multiple dynamic perturbations. Since we know the pattern of perturbations, we can better analyze the performance of SD arrangement and inversion in hemodynamic reconstruction.

This concern is addressed in lines 358-361 and 389-398 of the revised manuscript.

Point 2: Compared to the results presented in this paper in real imaging applications the forward solver used in the inversion procedure can show some deficiencies since the diffusion equation and the Born approximation have limitations and the background medium addressed is homogeneous differently from real media. The gap between the case here addressed and the real applications can have some consequences in the conclusions here formulated? I can understand that the MDMD arrangement can be anyway advantageous for the purpose of imaging applications, however some comments should be spent on this fact.

Response: Thanks. Based on our knowledge in this field and regarding the previous works, the changes in absorption (or scattering) related to brain activity are small with respect to the base-line values (1% to 5% for oxygenation, with the expected changes in scattering being even smaller) therefore the Born approximation can be used when describing the diffusion of light through brain tissue (Chiarelli et al., 2016). Whenever the intention is to measure hemodynamics from tissues other than the brain, then the Born approximation will not be accurate, and the interaction between the perturbations should be considered in the forward model (Sassaroli, Martelli, & Fantini, 2009).

Please refer to lines 389-398. And the results of Fig. 15.

Point 3: It is true that the authors present a comparison of the depth sensitivity of their analytical model with the Monte Carlo data on Colin 27 brain template. However, the results shown in Fig. 10 and 11 emphasizes difference that, although smaller for larger depth, are always present between analytical model and Monte Carlo regression. The authors simply conclude that Monte Carlo methods although accurate in the estimation of depth sensitivity are not well suited to be used in the forward model due to their long computation time. Is it possible to conclude that the differences observed are not decisive in the inversion procedure?

Response: Thanks for the insightful comment. The forward model in this manuscript is simple and one-layer; we have mentioned this in the discussion section. We could use FEM to solve the diffusion equation in a sophisticated and multi-layered Slab medium. Since the Analytical solution for sophisticated and anatomical mediums are not available. Based on the comment of reviwer#3, we have compared the computation time of FEM and Analytical solution when applied to the Diffusion Equation. Analytical simulation of 36 channels, including a sampling rate of 30 samples per second, takes less than 10 minutes. For the same sampling rate and channels, the FEM simulation will take almost 43-48 hours, depending on system speed. 

If we use an accurate forward model, then all this concern can be solved. It can be concluded that the results won't be decisive in the inversion procedure. With the help of insightful comments of reviewer #2, we have added instrumentational noise (Gaussian noise) to each channel, and then after inversion, the improvement obtained by MDMD has been preserved (please refer to the response of Point 2 of the second referee). 

In order to cover major concerns in points 1-3, we have added several paragraphs along with simulations. We have added uncertainties to elements of the Jacobian matrix with random distribution. Then we have simulated the forward model. The contrast, after considering the random behavior, is also simulated in Fig 1. The changes are visible.

Fig 1: The contrast (depth sensitivity) in forward model by adding random coefficients.

Please refer to lines 331-357. Still the performance of MDMD is superior than others.

The simulation results illustrated in Fig 15 of the revised manuscript:

Fig 15. (a) Represents Gaussian distribution (mean=1 and δ=0.07) of Rnd matrix elements. (b) Boxplot regarding random coefficients. (c) Represents Gaussian distribution (mean=1 and δ=0.2) of Rnd matrix elements. (d) Boxplot regarding random coefficients.

 Minor Points:

Point 1: At line 60 the single Ref. 15 is not enough, I would also add Ref. 16.

Response: Thanks. The reference was added to the revised version of the manuscript.

Point 2: At line 62 it is written “… simple slab boundary condition in semi-infinite geometry are utilized [16]”. The sentence is confusing in the sense that is confused the meaning of geometry and boundary condition. Here the term boundary condition appears misleading. To me would make sense to rephrase as “… the simple geometry, that can approximate a semi-infinite medium for thick slabs, is utilized [16]”.

Response: Thanks. Corrected.

Point3: At line 96 it is written “The analytical solution for perturbative DE has been obtained in the boundary condition of Fig. 3.” So far, I understand it should be “The analytical solution for perturbative DE has been obtained for the geometry of Fig. 3.” Boundary condition and geometry have a different meaning in this context. The actual boundary condition used to solve DE for the forward solver used in this paper is the extrapolated boundary condition that is not a geometry but a condition to make an energy balance at the external interface of the medium.

Response: Thanks. Corrected.

Point4: At line 102 the symbol S (The thickness of the slab) is not defined. NB that S is also the symbol used for the different regions S1-S9 (See Fig. 3).

Response: Thanks. Corrected “the thickness of slab is equal to 40mm” We have removed “S” to avoid confusion with region name.

Point5: At line 104 is written “Consequently, the volume of the inhomogeneity (inclusion) is regarded to be small concerning baseline optical properties of the homogeneous medium [16].” It should be “Consequently, the volume of the inhomogeneity (inclusion) is regarded to be small concerning baseline optical properties of the homogeneous medium [16].”

Response: Thanks Corrected.

Point6: At line 112 it is written “The final expression of Rpert(\\rho) for each channel derived based on Born approximation [21].” It looks like the sentence miss the final part. The sentence recalls the Born approximation; however, Ref. 21 is mainly related to higher order perturbation theory. Do the authors mean that in Ref. 21 the results for the Born approximation are also presented?

Response: Thanks. We have cited this ref because it has a comprehensive view of different theories including Born approximation. 

Point7: At line 136 the acronym HRF is used without definition. I understand that implicitly means Hemodynamic Response Function, however why it should be omitted this definition?

Response: Thanks. We have defined this acronym several lines before (line 128). In Fig.2 of “Response to Reviewers” we have illustrated HRF signal. The signal looks like hemodynamic response function that is why it is called HRF in (Bonomini et al., 2015) it is also called HRF. This reference is one our main source of study.

〖∆HbO_2〗_i (t)=〖HRF〗_i (t)*s(t)+ ∅_phy (t) and ∆〖Hb〗_i (t)=-1/3×〖∆HbO_2〗_i (t)

Fig.2: The procedure of Oxy and Deoxy hemoglobin generation.

Point8: At line 272, Eq. (15), index of the sum in Eq. (15) is a real number, while has not been used an index numbering the number of Source-Detector pairs?

Response: Thanks. Agreed. That parameter is changed in the revised manuscript to x_SD (line 278).

Point10: The actual title of Ref. 16 is: “Light Propagation through Biological Tissue and other Diffusive Media: Theory, Solutions and Software” 2009.

Response: Thanks. Corrected.

Point11: In the first row of Fig. 4 is missed the info on the y axis (Z (mm)?).

Response: Done.

Point12: In the second row of Fig. 6 is missed the info on the x axis (Time s?).

Response: Done.

Reviewer#2:

Comments:

Author report on a simulation platform for diffuse optical tomography (DOT) adapted for reconstruction of hemodynamic responses. The platform is based on analytical solutions, under the Born approximation, of the diffuse equation for a semi-infinite slab. The work is technically sound and, together with the claimed availability of software and data, will be useful for setting up DOT systems.

Response: We appreciate the positive feedback of the esteemed reviewer. Of course, the simulation code and a simulation guide will be available according to PLOS One policies. Both of them will help the other researcher to set up a system for their purposes.

 Major Concerns

Point 1: At my opinion there are the following points to address and clarify: Reconstruction: what I don’t understand is whether the reconstruction is forced at the depth of 15 mm or it is performed in the whole volume. In the first case, the point has to be emphasized and better specified in the text and, at my opinion, the method can’t be properly called “tomography”. In the second case, as well, it has to be emphasized in the text and, what I expect, is a figure representing slices in the volume at a defined time Ts to see the reconstructed depth that, in DOT, is typically underestimated.

Response: Thanks for the insightful comment of the reviewer. The reconstruction is forced at 15mm depth. However, by the simulation approach, it is possible to do tomography if we add other perturbations in different depth. Then it is required to generate a Jacobian matrix for each depth. We have corrected the “Tomography” to “Topography” in the manuscript.

Point 2: Noise and Inverse-crime: As far as I’ve understood, the only noise added is on the optical properties of the S5 and nearby voxels. This means that the forward model has been generated with well-defined optical properties using the Jacobian matrix, and the inversion has been performed using the same matrix on unnoisy data ΔR. This is typically called an “inverse-crime”. I suggest to add noise (Gaussian or Poisson) to the simulated data to avoid this problem.

Response: Thanks for the valuable comment. To avoid inverse-crime, we have added simulation and explanation of how noise affects the results (please refer to lines 309-330, Fig.13 and Fig.14):

Fig 13. Demonstrates the effect of noise on performance assessment of different SD arrangements in hemodynamic extraction of the S5 region concerning the regularization parameter.

Fig 14. (a) Represents the ability of hemodynamic extraction from the S5-region concerning several SNRs. (b) illustrates that in the worst case condition (SNR=32 dB), the optimum point for ϵ has changed from 〖10〗^(-5) to 〖10〗^(-4). (c) and (d) Represents reconstructed and synthetic ∆HbO_2 and ∆Hb for MDMD and HD-DOT respectively.

 Minor Points

Point 1: the acronym HRF is not specified in the text.

Response: Corrected. Please refer to line 128.

Point 2: Figure 1 is reported with a very poor resolution, writings are too small.

Response: Thanks. We have increased the writing font. The resolution has been improved.

Reviewer#3:

Comments:

The authors present a simulation setup for the performance assessment of different source-detector configurations in high density diffuse optical tomography. The information content in the paper is well organised. While the work presented is interesting and very useful research, a few concerns are raised below, which are to be addressed prior to any publication.

Response: We appreciate the positive feedback of the esteemed reviewer.

 Major Concerns

Point 1: One of the important aspects of the presented work is the significant reduction in computational time due to the use of analytical forward model, when compared to finite element and Monte Carlo methods. Therefore, it is crucial to provide the readers with the comparison of individual computational times for a sample forward model. Compare with the result of FEM

Response: Thanks for the insightful comment. We have provided a quantitative comparison between Analytical and FEM. Analytical simulation of 36 channels, including a sampling rate of 30 samples per second, takes less than 10 minutes. For the same sampling rate and channels, the FEM simulation (Fig 3 of “Response to Reviewers”) will take almost 43-48 hours, depending on system speed) We have excluded the computation time of 4 extra channels(. Please refer to lines 303-306.

Fig. 12: Geometry developed for FEM simulation of 40 channels

Fig. 3: ∆OD for 40 Channels in two wavelength

Point 2: From Figure 9, and Table-1, the authors want to show that higher the number of unique elements in Jacobian matrix, better will be the recovery. This can be misleading. While it is logical that a Jacobian with more independent information can give a better recovery, the correct way to observe this is to compare the normalised singular values of different Jacobian matrices corresponding to the SD arrangements. I would recommend looking into: Optics Letters Vol. 26, Issue 10, pp. 701-703 (2001).

Response: Thanks for the comment. SDV analysis is applied to the Jacobian matrix of different SD arrangement:

A singular value analysis of the Jacobian matrix associated with introduced SD arrangements is used as a benchmark (Culver, Ntziachristos, Holboke, & Yodh, 2001). Besides unique elements of the Jacobian matrix, the singular value analysis of the different methods in Fig 10 indicates that both shape of the singular value spectra and the magnitude for MDMD arrangement is higher than other SD combinations. Please refer to lines 253-256.

Fig 10. Singular value spectra for Jacobian matrix of HD-DOT, MDMD, Arrangement-1 and 2

Point 3: Figure 10 and 11 have same information, therefore figure 10 can be avoided and two additional sub-plots for SD separation 10 and 20mm can be included in Figure 11.

Response: Done. Fig. 10 has been avoided and Fig. 11 in the manuscript represents depth sensitivity of Analytical and MC including SD=10mm and SD=20mm.

Fig 11. Represents the depth sensitivity of the analytical pDE in Slab medium and depth sensitivity of the regression of MC on Colin 27 geometry for SD separation of 10mm-60mm. 

 Minor Points

Point 1: Add reference for lines 150-152.

Response: Thanks. The Reference is added.

Point 2: Use one abbreviation for oxy hemoglobin (either O2Hb or HbO2), and for deoxy hemoglobin (either HHb or Hb) throughout the manuscript.

Response: Thanks. Corrected.

Point 3: Important future directions such as the use of multi-layered medium, fit better in the conclusion section rather than discussions. Re-edit the conclusion section to incorporate this information seamlessly.

Response: Thanks. The future directions are removed from discussion section and moved to conclusion as follow:

Based on the modeling approach and simulation results, the achievements of the MDMD looks more beneficial than other methods. But still, there would be a gap between the outcomes of this study and real imaging applications. It is suggested to use the proposed simulation approach with a modified forward model according to the following suggestions. The brain model in this work is one-layer, while the multi-layered medium can be considered to get closer to the real results. Also, we must find the interaction between the perturbations in the solution of DE equations (whenever the perturbations are not small regarding the baseline optical properties), while Born approximation is used in this work. In this study the diffuse reflectance modulated due to change in absorption coefficient. The optical scattering coefficient changes along with the absorption coefficient should be taken into account for more accurate simulation of diffuse reflectance. Please refer to lines 389-398.

Reviewer#4:

Comments:

 Major Concerns:

Point 1: In the manuscript, ‘Performance Assessment of High-Density Diffuse Optical Tomography Regarding Source-Detector Array Topology,’ the authors present a simulation study comparing various source-detector separation distances. While the methods are sound, the choice of methods (Born approximation, limited number of source-detector pairs and therefore a small field of view, analytical model instead of anatomy-based FEM using either diffusion or Monte Carlo, the single layer of optical properties) leads to the results and discussion providing limited information to the community for further advances in simulation, system design, or empirical considerations. As such, the authors are encouraged to add complexity to their modeling procedures or better motivate their choices to place their work in context with the current status of the field.

Response: Thanks for the positive feedback and concerns. According to the response to comments of esteemed referees. We have added complexity according to the reviewer’s suggestions:

 The instrumentational Noise is added to each channel. The effect of Noise on the previous results are illustrated in Fig13 and Fig 14.

 We have added uncertainties to elements of the Jacobian matrix with random distribution. The simulation results can be found in Fig 15 of the revised manuscript.

 We have compared the computation time of FEM and Analytical methods quantitatively. Please refer to lines 303-306.

We have mentioned that why we could use Born approximation (please refer to the response to respected reviewer#1). Of course, the number of SDs is limited, but it can be expanded. Increasing the number of SDs to cover a large area of the brain will increase the computation time. But it worth to expand it. However, for the first step of the study, it is better to focus on a small area and discuss everything in detail. Imagine each SD arrangement as a single probe; if we sweep this probe spatially on the brain surface, then we will get the results of expansion, and we can have an expanded field of view.

If we want to use sophisticated slab medium or anatomical medium because of complicated geometry, the analytical solutions are not available (Strangman, Li, & Zhang, 2013). However, we are able to use FEM when it is applied to the Diffusion Equation (Okada & Delpy, 2003).

We do not recommend the use of Monte Carlo as a forward model in this simulation setup. MC can be used to study depth sensitivity in a real brain template for one-time research. However, if we add dynamic perturbation inside the brain template, we need to run, for example, 4000 run for each sampling rate. Instead of MC analysis, we recommend doing an experiment in tissue-like phantom. If we apply different arrangements to tissue-like phantom (as a forward model), there may be a chance to produce synthetic hemodynamic, but it is still challenging. The challenge is how to insert dynamic perturbations inside the phantom.

Authors Modification:

Fig 2 has been modified to better match with the simulation codes. 

References

Bonomini, V., Zucchelli, L., Re, R., Ieva, F., Spinelli, L., Contini, D., … Torricelli, A. (2015). Linear regression models and k-means clustering for statistical analysis of fNIRS data. Biomedical Optics Express, 6(2), 615. https://doi.org/10.1364/BOE.6.000615

Chiarelli, A. M., Maclin, E. L., Low, K. A., Mathewson, K. E., Fabiani, M., & Gratton, G. (2016). Combining energy and Laplacian regularization to accurately retrieve the depth of brain activity of diffuse optical tomographic data. Journal of Biomedical Optics, 21(3), 036008. https://doi.org/10.1117/1.JBO.21.3.036008

Culver, J. P., Ntziachristos, V., Holboke, M. J., & Yodh, A. G. (2001). Optimization of optode arrangements for diffuse optical tomography: A singular-value analysis. Optics Letters, 26(10), 701–703.

Martelli, F., Del Bianco, S., & Zaccanti, G. (2005). Perturbation model for light propagation through diffusive layered media. Physics in Medicine & Biology, 50(9), 2159.

Okada, E., & Delpy, D. T. (2003). Near-infrared light propagation in an adult head model I Modeling of low-level scattering in the cerebrospinal fluid layer. Applied Optics, 42(16), 2906. https://doi.org/10.1364/AO.42.002906

Sassaroli, A., Martelli, F., & Fantini, S. (2009). Higher-order perturbation theory for the diffusion equation in heterogeneous media: application to layered and slab geometries. Applied Optics, 48(10), D62-73. https://doi.org/10.1364/AO.48.000D62

Strangman, G. E., Li, Z., & Zhang, Q. (2013). Depth Sensitivity and Source-Detector Separations for Near Infrared Spectroscopy Based on the Colin27 Brain Template. 8(8). https://doi.org/10.1371/journal.pone.0066319

---

## [Decision Letter · Decision Letter 1]

25 Feb 2020

Performance Assessment of High-Density Diffuse Optical Topography Regarding Source-Detector Array Topology

PONE-D-19-31493R1

Dear Dr. Setarehdan,

We are pleased to inform you that your manuscript has been judged scientifically suitable for publication and will be formally accepted for publication once it complies with all outstanding technical requirements.

With kind regards,

Alberto Dalla Mora, Ph.D.

Academic Editor

PLOS ONE

Additional Editor Comments (optional):

Reviewers' comments:

Reviewer's Responses to Questions

**Comments to the Author**

1. If the authors have adequately addressed your comments raised in a previous round of review and you feel that this manuscript is now acceptable for publication, you may indicate that here to bypass the “Comments to the Author” section, enter your conflict of interest statement in the “Confidential to Editor” section, and submit your "Accept" recommendation.

Reviewer #1: All comments have been addressed

Reviewer #2: All comments have been addressed

Reviewer #4: All comments have been addressed

2. Is the manuscript technically sound, and do the data support the conclusions?

Reviewer #1: Yes

Reviewer #2: Yes

Reviewer #4: Yes

3. Has the statistical analysis been performed appropriately and rigorously? 

Reviewer #1: Yes

Reviewer #2: Yes

Reviewer #4: Yes

4. Have the authors made all data underlying the findings in their manuscript fully available?

Reviewer #1: Yes

Reviewer #2: No

Reviewer #4: Yes

5. Is the manuscript presented in an intelligible fashion and written in standard English?

Reviewer #1: Yes

Reviewer #2: Yes

Reviewer #4: Yes

6. Review Comments to the Author

Reviewer #1: I have read carefully the authors response to my reports and to all the reports of the other reviewers together with the changes introduced in the revised version of the manuscript. Indeed, the manuscript has been significantly improved compared the previous version and the main points raised in the reports has been properly addressed. According to this fact, I recommend its publication on Plos One unaltered.

Reviewer #2: (No Response)

Reviewer #4: The authors have greatly improved the manuscript. The manuscript is now appropriate for this journal.

7. PLOS authors have the option to publish the peer review history of their article (what does this mean?). If published, this will include your full peer review and any attached files.

Reviewer #1: No

Reviewer #2: No

Reviewer #4: No

---

## [Editor Report · Acceptance letter]

9 Mar 2020

PONE-D-19-31493R1 

Performance Assessment of High-Density Diffuse Optical Topography Regarding Source-Detector Array Topology 

Dear Dr. Setarehdan:

I am pleased to inform you that your manuscript has been deemed suitable for publication in PLOS ONE. Congratulations! Your manuscript is now with our production department. 

With kind regards,

on behalf of

Professor Alberto Dalla Mora 

Academic Editor

PLOS ONE